# A multiplexed siRNA screen identifies key kinase signaling networks of brain glia

Jong-Heon Kim[1], Jin Han[2], Ruqayya Afridi[2], Jae-Hong Kim[1], Md Habibur Rahman[1,6], Dong Ho Park[3], Won Suk Lee[4], Gyun Jee Song[5], Kyoungho Suk[1,2]

The dynamic behaviors of brain glial cells in various neuro-inflammatory conditions and neurological disorders have been reported; however, little is known about the underlying intracellular signaling pathways. Here, we developed a multiplexed kinome-wide siRNA screen to identify the kinases regulating several inflammatory phenotypes of mouse glial cells in culture, including inflammatory activation, migration, and phagocytosis of glia. Subsequent proof-of-concept experiments involving genetic and pharmacological inhibitions indicated the importance of T-cell receptor signaling components in microglial activation and a metabolic shift from glycolysis to oxidative phosphorylation in astrocyte migration. This time- and cost-effective multiplexed kinome siRNA screen efficiently provides exploitable drug targets and novel insight into the mechanisms underlying the phenotypic regulation of glial cells and neuroinflammation. Moreover, the kinases identified in this screen may be relevant in other inflammatory diseases and cancer, wherein kinases play a critical role in disease signaling pathways.

## Introduction

The central nervous system (CNS) includes a sophisticated immune system that actively participates in maintaining homeostasis and resolving infections. Glial cells, such as astrocytes and microglia, are nonneuronal cells that play key roles in homeostasis and immune surveillance (Engelhardt et al, 2017). Because these cells undergo extensive morphological and functional changes during neuroinflammation, they are commonly termed as reactive or activated glia (Liddelow & Barres, 2017). However, such terminology is currently controversial (Escartin et al, 2019); thus, the neuro-inflammatory aspects of glial responses require further investigation and a better definition. Nevertheless, there is a consensus on the significant changes in glial phenotypes in response to neuroinflammatory or pathological stimuli. In fact, accumulating evidence has revealed a spectrum of glial phenotypes, including cellular (re)activation, proliferation, migration, and phagocytosis (Burda & Sofroniew, 2014). These cellular phenotypes are primarily involved in resolving encountered challenges and promote tissue remodeling. For instance, after an acute focal injury, activated microglia and astrocytes produce cytokines to propagate inflammatory signals. The signals released by activated glia attract other glial cells to the injury site, leading to the clearance of cellular debris or pathogenic molecules, reconstruction of tissues, and reestablishment of homeostasis. However, in cases of severe or chronic neuroinflammation, these phenotypes can cause detrimental effects, such as neurodegeneration and CNS tissue damage. Indeed, activated glial cells release neurotoxic molecules, such as reactive oxygen and nitrogen species (Kempuraj et al, 2017), interfere with neuronal activities (Robel & Sontheimer, 2016) and cause nonspecific phagocytosis (Jung & Chung, 2018). The response of activated glia to CNS damage involves a complex combination of events that should ideally be balanced between beneficial and detrimental effects. Inappropriate glial cell activity is a "common denominator" in a wide spectrum of neurological disorders (Kempuraj et al, 2017); thus, elucidating the molecular mechanisms underlying glial activation could facilitate the discovery and development of viable therapeutic targets. However, little is known about the intracellular signaling pathways underlying the diverse phenotypes and associated behaviors of glial cells.

Kinases comprise a large group of enzymes that not only regulate protein function by phosphorylating serine, threonine or tyrosine residues on target proteins but also contain a regulatory domain for other kinases. Kinases play a predominant regulatory role in nearly all cellular processes, including cell proliferation, differentiation, growth, metabolism, cytoskeletal rearrangements, and survival. Therefore, exploring the role of kinases is essential for understanding the modulation of glial phenotypes or their hyperactivity. Aberrant kinase activity has been the focus of molecular research

[1]Brain Science & Engineering Institute, Kyungpook National University, Daegu, Republic of Korea   [2]Department of Biomedical Science, Department of Pharmacology, School of Medicine, Kyungpook National University, Daegu, Republic of Korea   [3]Department of Ophthalmology, School of Medicine, Kyungpook National University, Kyungpook National University Hospital, Daegu, Republic of Korea   [4]Neuracle Science Co., Ltd. Seoul, Republic Korea   [5]Department of Medical Science, College of Medicine, Catholic Kwandong University, Gangneung-si, Republic Korea; Translational Brain Research Center, International St. Mary's Hospital, Catholic Kwandong University, Incheon, Republic Korea   [6]Department of Neurology, The Johns Hopkins University School of Medicine, Baltimore, MD, USA

Correspondence: ksuk@knu.ac.kr

on various diseases, particularly those involving inflammation or malignant proliferation, such as rheumatoid arthritis or cancer. As such, the kinase-targeting studies published to date have primarily focused on nonCNS diseases (Roskoski, 2019). Specific kinases, including neuronal leucine-rich repeat kinase 2 and glycogen synthase kinase-3$\beta$ in Parkinson's disease (Li et al, 2014), and τ tubulin kinase 1 in Alzheimer's disease and other neurodegenerative disorders (Nozal & Martinez, 2019), were targeted in such studies. However, to date, research on glial kinases has been limited. Therefore, in this study, we present a novel approach for identifying kinases with important roles in the regulation of neuroinflammatory glial phenotypes and behaviors, such as glial activation, death/survival, migration, and phagocytosis. To simultaneously address various glial cell behaviors, we designed and performed a multiplexed cell-based screen of an siRNA library targeting 623 kinases in LPS-stimulated mixed glial cell (MGC) cultures prepared using the brains of neonatal mice. Four different phenotypic assays were performed in parallel or tandem sequence. Using stringent criteria, we identified hit kinases and selected some for further validation and characterization. Our findings provide insight into the largely unknown intracellular signaling networks of glia controlling neuroinflammation and highlight the therapeutic potential of targeting glial kinases to modulate glial phenotypes and treat neuroinflammatory disorders.

## Results

### Multiplexed kinase siRNA screen in brain glial cells

To identify kinases that modulate the inflammatory activation of glia and their function during neuroinflammation, we performed a multiplexed kinome-wide siRNA screen in primary MGC cultures stimulated with the proinflammatory endotoxin LPS (Fig 1). The kinome-wide siRNA screen was performed in a 96-well plate format. We used an siRNA library containing a pool of three siRNAs targeting each of the 623 kinases across seven plates. The seven library plates were reformatted into 24 plates for the assay to obtain data in triplicate (by splitting a single pool of kinase siRNAs into three wells). MGCs, obtained from the brains of 20 pups for a single screening round, were seeded into the assay plates at a density of 2,500 cells per well and transfected with the siRNA library, non-targeting siRNA or transfection reagent alone. MGCs were then incubated with or without LPS (1 $\mu$g/ml) for 48 h (to a total of 48 plates; 24 LPS-treated and 24 LPS-untreated plates). After LPS stimulation, four different assays were performed using either the culture medium or the remaining cells in the assay plates. All measurements were performed in triplicate. The whole multiplexed siRNA screen was repeated twice (96 plates in total). The LPS-untreated condition was used as screening control. Quality checks across different plates or culture conditions were performed by including multiple controls, such as LPS-untreated MGCs, non-targeting siRNA-transfected cells, and transfection reagent alone, in every assay plate. To determine the cellular proportions in different MGC preparations, we performed immunocytochemical staining with glia-specific markers (Fig S1).

Four different phenotypes were examined in parallel or tandem sequence in a multiplexed manner: (i) inflammatory glial activation (via a nitric oxide [NO] assay); (ii) glial cell death/survival (via a cytotoxicity assay); (iii) glial migration (via a wound healing assay); and (iv) glial phagocytosis (via a Zymosan uptake assay; Fig 1). The sequence in the multiplexed screen was simulated in vivo conditions, in which glia are first activated, then they migrate, and finally perform phagocytosis at the site of brain inflammation. Moreover, the multiplexed screen allowed multiple assays to be conducted under the same conditions, thereby reducing experimental variation and saving time and reagents. The reproducibility of all assays was confirmed by at least duplicate siRNA screens. A Pearson's correlation coefficients for correlations between replicate screens were >0.8, considered to be a strong correlation as previously described (Mukaka, 2012): NO assay (r = 0.8134), wound healing assay (r = 0.8872), cytotoxicity assay (r = 0.8893), and Zymosan uptake assay (r = 0.8744). To identify the kinases engaged in glial phenotype regulation in an LPS-dependent manner, we compared the hits identified in the presence and absence of LPS stimulation (Fig S2). In the wound healing and Zymosan uptake assays, 45 and 31 hit kinases were LPS-dependent, respectively (Table S1). These data were used as baseline reference for determining the most relevant hits.

The efficiency of the siRNA-mediated knockdown of kinase gene expression was validated using real-time PCR and immunoblotting for the selected kinases. For validation at the mRNA level, 12 kinases were selected based on phenotypic screening of the siRNA library (Table S2); four from the activation screen; four from the migration screen; three from the phagocytosis screen; and four that did not show significant effects in any screens (Fig S3A). For validation at the protein level, six kinases were selected based on inhibition efficiency at the mRNA level (>50%, three kinases; <50%, three kinases; Fig S3B). Our data showed similar siRNA down-regulation of specific kinases (mRNA/protein) in PBS and LPS conditions: all reduction rates (%) had a $P$-value >0.1 in comparisons between PBS and LPS conditions.

After immunocytochemical staining, the analysis of fluorescent images revealed the cellular composition of cultured MGCs (Fig S1 and Table S3). MGCs comprised approximately 66% astrocytes, 15% microglia, and 15% oligodendrocytes in the LPS-untreated group and 64% astrocytes, 23% microglia, and 13% oligodendrocytes in the LPS-treated group. The interassay variation was <10% across different preparations or <20% between LPS-untreated and LPS-treated wells, indicating statistical consistency (Li et al, 2017). In this study, we focused on the phenotypes of astrocytes and microglia—the main players in neuroinflammation. The efficiency of siRNA transfection was determined using a Cy3-labeled siRNA transfection control (siRNA of GAPDH) followed by immunocytochemical staining with antibodies against glial cell markers, such as anti-glial fibrillary acidic protein (GFAP) or anti-ionized calcium binding protein (Iba-1; Fig S3C). Transfection rates of astrocytes and microglia (70.1% and 54.8%, respectively) in the LPS-untreated group and of those (67.1% and 56.3%, respectively) in the LPS-treated group were determined, and the intergroup variation in transfection rate was found to be <20% (Table S3). Therefore, both glial cell types were efficiently transfected with siRNAs; similarly, the kinase expression in astrocytes and microglia was successfully knocked down in the MGC model; thus, the observed phenotypic changes are likely relevant to both glial cell types.

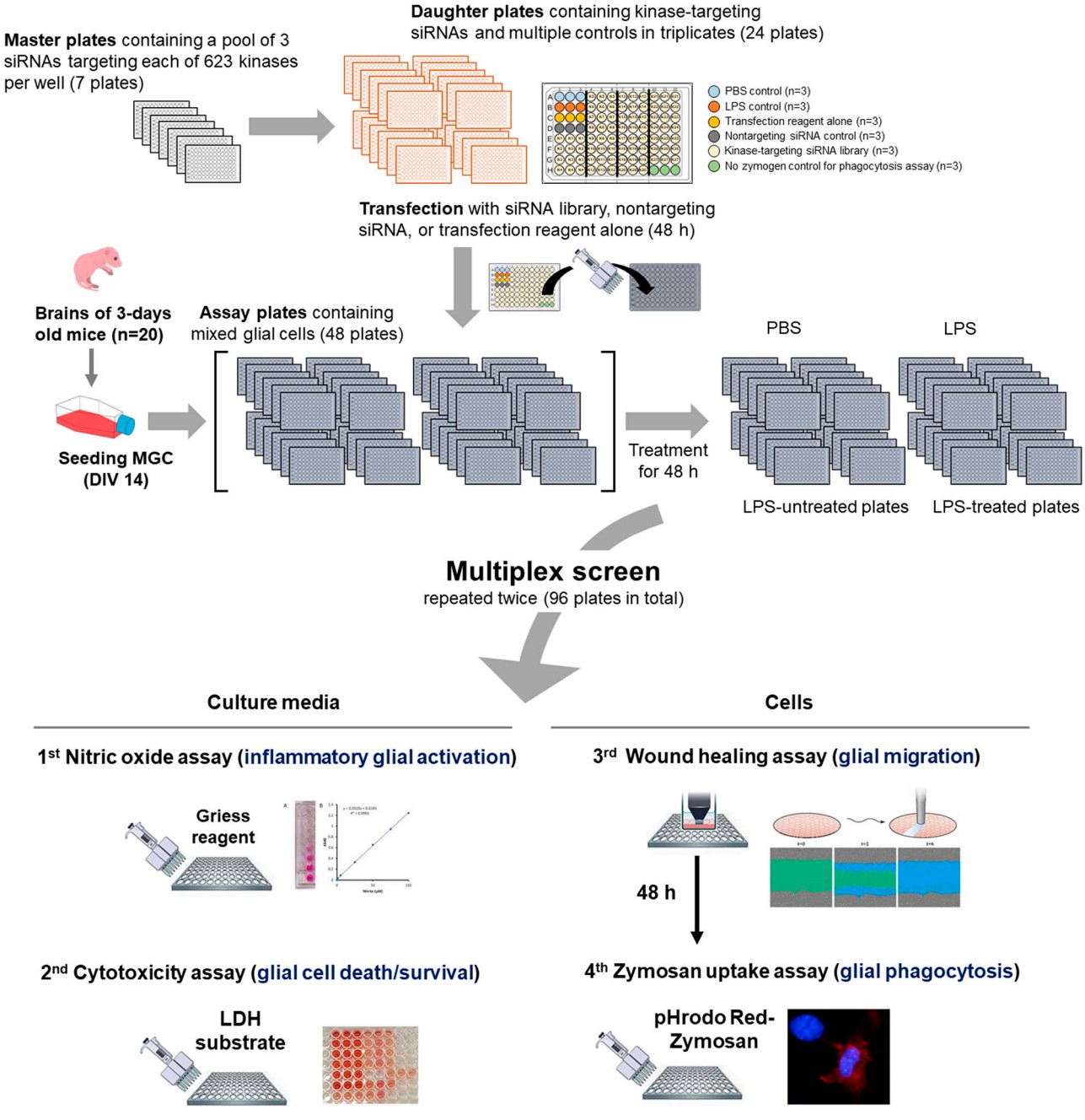

**Figure 1. Overview of the multiplexed kinase siRNA screening strategy.**
For the kinome-wide siRNA screening, we used the siRNA mouse kinase library in a 96-well plate format. Each well contained a single pool of three different siRNA sequences targeting each of the 623 kinases (seven plates). The seven library plates (master plates) were reformatted to 24 plates (daughter plates) to conduct the assay and obtain data in triplicate. Mixed glial cells (MGCs) were seeded into 24 plates (assay plates) at a density of 2,500 cells per well and transfected with the siRNA library, nontargeting siRNA or the transfection reagent alone. MGCs were then incubated with or without LPS (1 μg/ml) for 48 h (to a total of 48 plates with 24 LPS-treated plates and 24 LPS-untreated plates). After LPS stimulation, four different assays were performed using either the culture medium or the remaining cells in the assay plates. All measurements were performed in triplicate. The whole multiplexed siRNA screen was repeated twice (96 plates in total). For a single screening round, MGCs were prepared from the brains of 20 pups. The LPS-untreated condition was used as screening control. See the "Materials and Methods" section for detailed descriptions of individual assays.

### The screen identifies glial kinases that regulate inflammatory activation of glia

After siRNA-mediated knockdown and LPS treatment of MGC in 96-well plates, culture media were transferred into two 96-well plates. One plate was used for a NO assay to assess glial cell activation and the other for a cell toxicity assay (lactate dehydrogenase [LDH] assay). The remaining MGCs in the original plates were sequentially subjected to wound healing and phagocytosis assays using Zymosan particles (Fig 1).

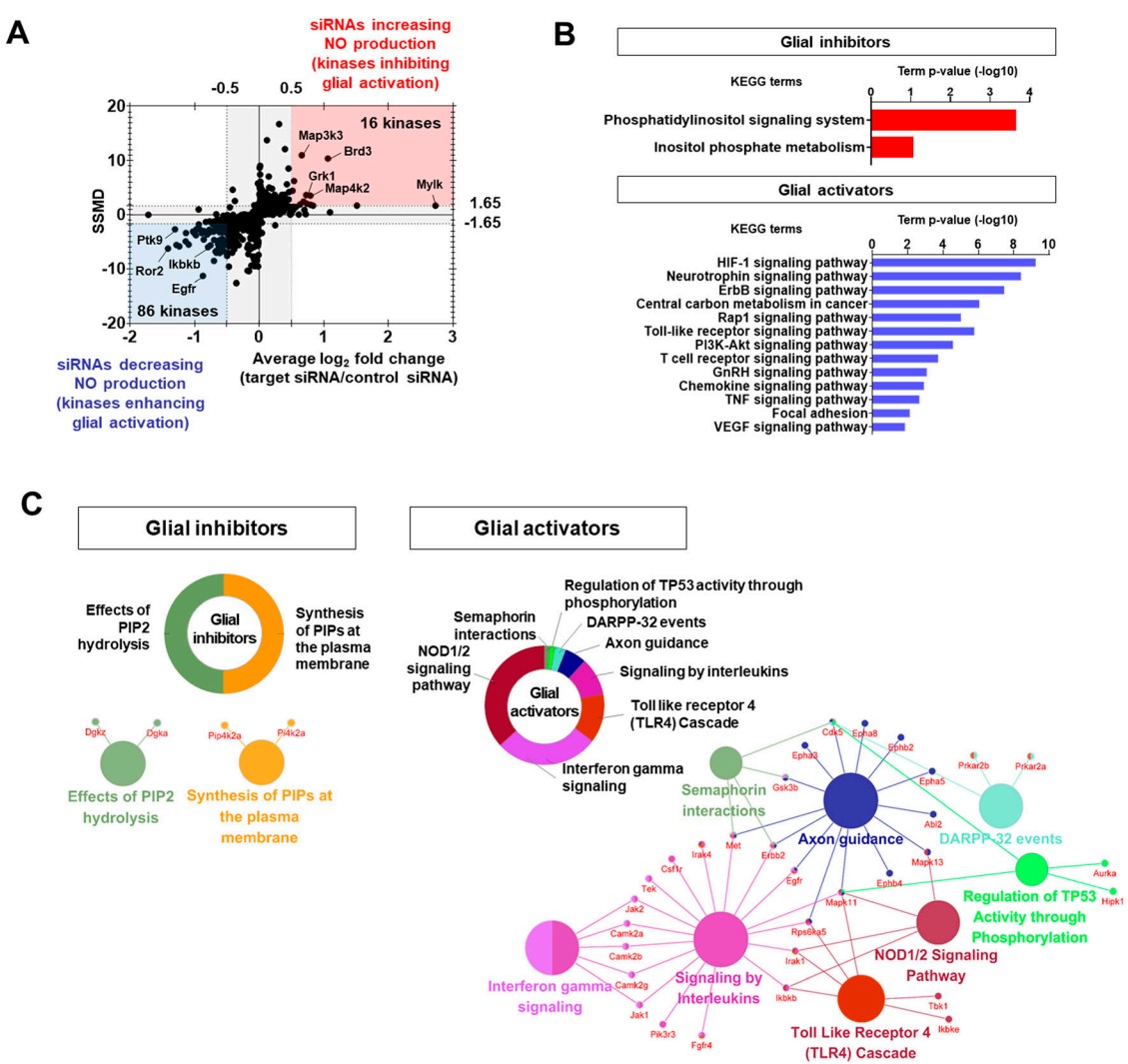

**Figure 2.  Kinases regulating glial activation.**
**(A)** Dual-flashlight plot for strictly standardized mean difference versus fold change (log$_2$ scale) in the nitric oxide assay. Increased or decreased phenotypes were defined by fold changes ≥0.5 or ≤−0.5, respectively (log$_2$ scale). Values on the x-axis indicate average fold change of three siRNAs for each target kinase. Final hit kinases were selected by strictly standardized mean difference values ≥1.65 or ≤−1.65. **(B)** Representative KEGG pathway analysis using the DAVID bioinformatics tool. **(C)** Representative functional group network views for Gene Ontology (GO) terms generated using ClueGO. Terms are functionally grouped based on shared genes (κ score) and shown in different colors.

NO production was used as an indicator of inflammatory activation of glia in this study, as in previous studies (Calabrese et al, 2007). In the first round of assays, NO production was assessed using the Griess assay, which measures nitrite (NO$_2^-$), a derivative of NO. Hits were selected using a dual-flashlight plot in which both the average fold change (FC) and the strictly standardized mean difference (SSMD) were considered simultaneously (Fig 2A). The following criteria were used to select siRNA hits: for siRNAs that increased NO production (kinases inhibiting glial activation), an average FC ≥ 0.5 (on the log$_2$ scale) and SSMD ≥1.65 were used; conversely, for siRNAs that decreased NO production (kinases enhancing glial activation), an average FC ≤ −0.5 (on the log$_2$ scale) and SSMD ≤ −1.65 were used. Overall, we identified 16 and 86 kinases that inhibited (glial inhibitors) or enhanced glial activation (glial activators), respectively (Fig 2A and Table S4). Using the Reactome

database (https://reactome.org/), kinase hits were compared with the "TLR4 signal pathway" including NF-kB, MAP kinase, and interferon signaling, which were expected hits in the screen (Fig S4A). Several TLR4 signaling-associated kinases, including Ikbkb (log$_2$FC = −0.8), Ikbke (log$_2$FC = −0.58), Mapk11 (log$_2$FC = −0.5), Rps6ka5 (log$_2$FC = −0.6), and Tbk1 (log$_2$FC = −0.8), were identified as significant regulators of NO production.

Next, the effects of siRNAs on NO production were pharmacologically confirmed by reassaying with commercially available small-molecule inhibitors for 24 kinases (Fig S4B). LPS-induced NO production was reduced by the inhibitors of glial activation-enhancing kinases such as Itk, Mlkl, Irak4, Ulk1, Pkc, Dgk, Ephb4, Src, Ikbkb, and Mapk14. Because of the limited number of NO-decreasing kinases identified in the screen (15.7%), the selected hits were coincidentally either NO-increasing kinases or kinases

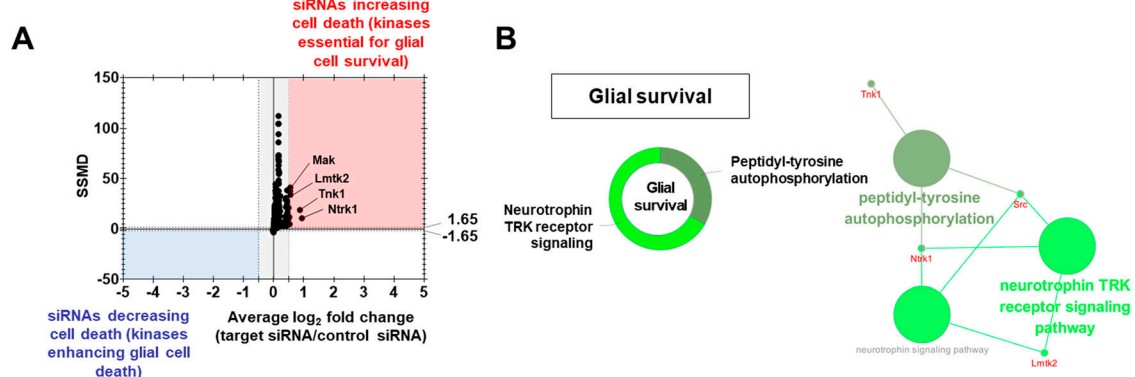

**Figure 3. Kinases regulating glial cell death.**
**(A)** Dual-flashlight plot for strictly standardized mean difference versus fold change (log$_2$ scale) in the cytotoxicity assay. Increased or decreased phenotypes were defined by fold changes ≥0.5 or ≤−0.5, respectively (log$_2$ scale). Values on the x-axis indicate the average fold change of the three siRNAs for each target kinase. The selected hit kinases had strictly standardized mean difference values ≥1.65 or ≤−1.65. Top kinases are indicated as well. **(B)** Representative functional group network view for Gene Ontology (GO) terms generated using ClueGO. Terms were functionally grouped based on shared genes (κ score) and shown in different colors.

without significant effects. These results were consistent with those obtained in the siRNA screen. The overall false positive rate was 25%, relatively low according to previous research (Mukaka, 2012). The effect size (Hedges' *g*) was also compared between the siRNA screen and pharmacological validation experiments; a *g* value ≥0.8 reflects a large effect size, whereas 0.5 and 0.2 can be considered moderate and small effect sizes, respectively (Jacob, 1998). Here, the average effect sizes for the original siRNA screen and pharmacological validation experiments were 2.96 and 4.42, respectively.

The 16 glia-inhibiting and 86 glia-activating kinases were subjected to enrichment analyses using Gene Ontology (GO) through the Database for Annotation, Visualization and Integrated Discovery (DAVID) (i.e., gene function classification) and Kyoto Encyclopedia of Genes and Genomes (KEGG) pathways and ClueGO (GO integration; Fig 2B and C). Among glia-inhibiting kinases, the most significantly enriched KEGG pathways were related to "phosphatidylinositol signaling system," and "inositol phosphate metabolism"; among the glia-activating kinases, the "HIF-1 signaling pathway," "TLR signaling pathway," and "chemokine signaling pathway," were the most significantly enriched (Fig 2B and Table S5). ClueGO analysis showed a significant enrichment of reactome pathways such as "effects of PIP2 hydrolysis" and "synthesis of PIPs at the plasma membrane" for glia-inhibiting kinases. However, "interferon γ signaling," "NOD1/2 signaling pathway," "semaphoring interactions," and "regulation of TP53 activity through phosphorylation" were highly enriched for glia-activating kinases (Fig 2C).

### Kinases associated with glial cell death/survival regulation

The second round of assays (cell toxicity assay) identified 15 kinases essential to glial cell viability (Fig 3A and Table S6). Hits were selected using a dual-flashlight plot in which both the average FC and SSMD were considered simultaneously (Fig 3A). To select siRNA hits that increase cell death (kinases essential for survival), we used the following criteria: an average FC ≥ 0.5 (on the log$_2$ scale) and SSMD ≥ 1.65. To select siRNA hits decreasing cell death (kinases enhancing

glial cell death), we used an average FC ≥ −0.5 (on the log$_2$ scale) and SSMD ≥ −1.65. Then, pharmacological validation was performed using commercially available small-molecule inhibitors for three kinases (Fig S5). Among them, Src kinase inhibitor (PP2) at 10 *μ*M enhanced cell death (Fig S5). Furthermore, ClueGO analysis indicated that "neurotrophin TRK receptor signaling pathway" and "peptidyl-tyrosine autophosphorylation" were significantly enriched in the 15 kinases (Fig 3B). The most significant KEGG pathways enriched for glial survival kinases were related to the "thyroid hormone signaling pathway," "inflammatory mediator regulation of TRP channels," "MAPK signaling pathway," and "Hh signaling pathway" (Table S7).

### Kinases controlling glial migration

In the third round of assays, we identified novel kinases regulating glial migration. Hits were selected using a dual-flashlight plot in which both the average FC and SSMD were considered simultaneously (Fig 4A). To select siRNA hits that accelerated migration (kinases that decelerated glial migration), we used the following criteria: an average FC ≥ 0.5 (on the log$_2$ scale) and SSMD ≥ 1.65; conversely, to select siRNA hits that decelerated migration (kinases that accelerated glial migration), we used an average FC ≤ −0.5 (on the log$_2$ scale) and SSMD ≤ −1.65. In the assay, 75 and 22 kinases that respectively decelerated (glial migration decelerators) or accelerated (glial migration accelerators) glial migration were identified (Fig 4A and Table S8). Subsequently, 24 kinase hits were pharmacologically validated by reassaying with small-molecule kinase inhibitors (Fig S6). Specific kinase inhibitors for Ulk1, Src, Ikbkb, Akt1, and Mapk14 decelerated glial migration, whereas inhibitors for Pdk2, Irak4, Frap1, Pkc, Pkam Pkm2, Ephb4, and Map2k5 accelerated glial migration. These data were generally in agreement with the siRNA screen (false-positive rate = 16%). The average effect sizes for the original siRNA screen and pharmacological validation experiments were 1.44 and 1.55, respectively.

According to KEGG pathway and GO enrichment analyses, the glial migration decelerators were related to "central carbon

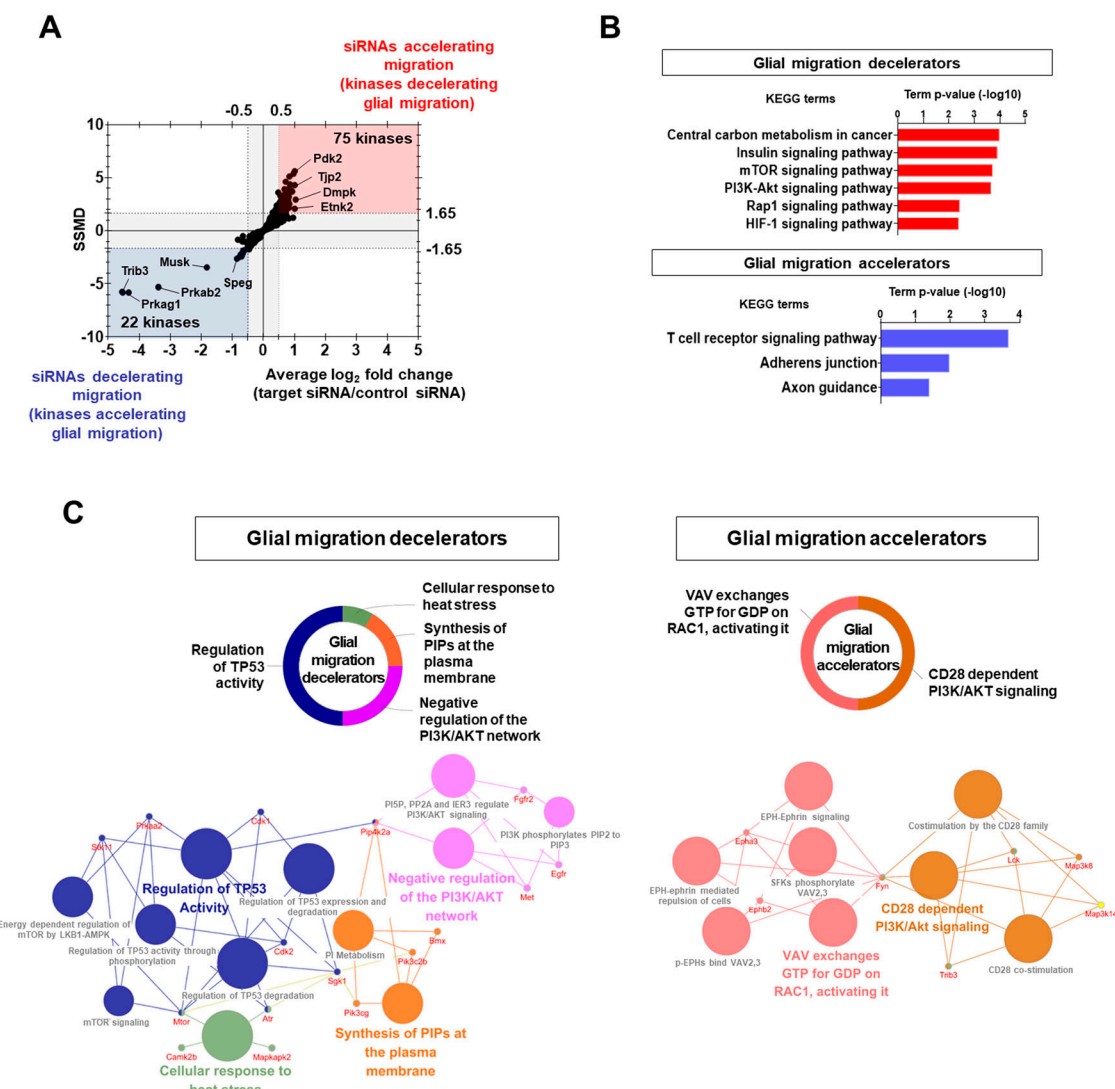

**Figure 4. Kinases regulating glial migration.**
**(A)** Dual-flashlight plot for strictly standardized mean difference versus fold change (log₂ scale) in the wound healing assay. Increased or decreased phenotypes were defined by fold changes ≥0.5 or ≤−0.5, respectively (log₂ scale). Values on the x-axis indicate the average fold change of three siRNAs for each target kinase. Final hit kinases were selected by strictly standardized mean difference values ≥1.65 or ≤−1.65. **(B)** Representative KEGG pathway analysis using the DAVID bioinformatics tool. **(C)** Representative functional group network views for Gene Ontology (GO) terms generated using ClueGO. Terms are functionally grouped based on shared genes (κ score) and shown in different colors.

metabolism in cancer," "insulin signaling pathway," "mTOR signaling pathway," "PI3K-Akt signaling pathway," "Rap1 signaling pathway," and "HIF-1 signaling pathway," whereas the glial migration accelerators were related to "TCR signaling pathway," "adherens junction," and "axon guidance" (Fig 4B and Table S9). ClueGO analysis revealed that "regulation of TP53 activity," "negative regulation of the PI3K/AKT network," "synthesis of PIPs at the plasma membrane," and "cellular response to heat stress" were significant reactome pathways for the glial migration decelerators; reactome pathways such as "VAV exchanges GTP for GDP on RAC1, activating it" and "CD28 dependent PI3/AKT signaling" were associated with glial migration accelerators (Fig 4C).

**Kinases regulating glial phagocytosis**

In the final round of assays, we identified 57 kinases that inhibited phagocytosis (siRNAs increased phagocytosis) and 124 kinases that enhanced phagocytosis (siRNAs decreased phagocytosis; Fig 5A and Table S10). Hit selection was performed using a dual-flashlight plot in which both average FC and SSMD were considered simultaneously. To select siRNA hits that increased phagocytosis (kinases that inhibited phagocytosis), we used the following criteria: an average FC ≥ 0.5 (on the log₂ scale) and SSMD ≥ 1.65; conversely, to select siRNA hits that decreased phagocytosis (kinases that enhanced phagocytosis), we used an average FC ≤ −0.5 (on the log₂ scale) and SSMD ≤ −1.65.

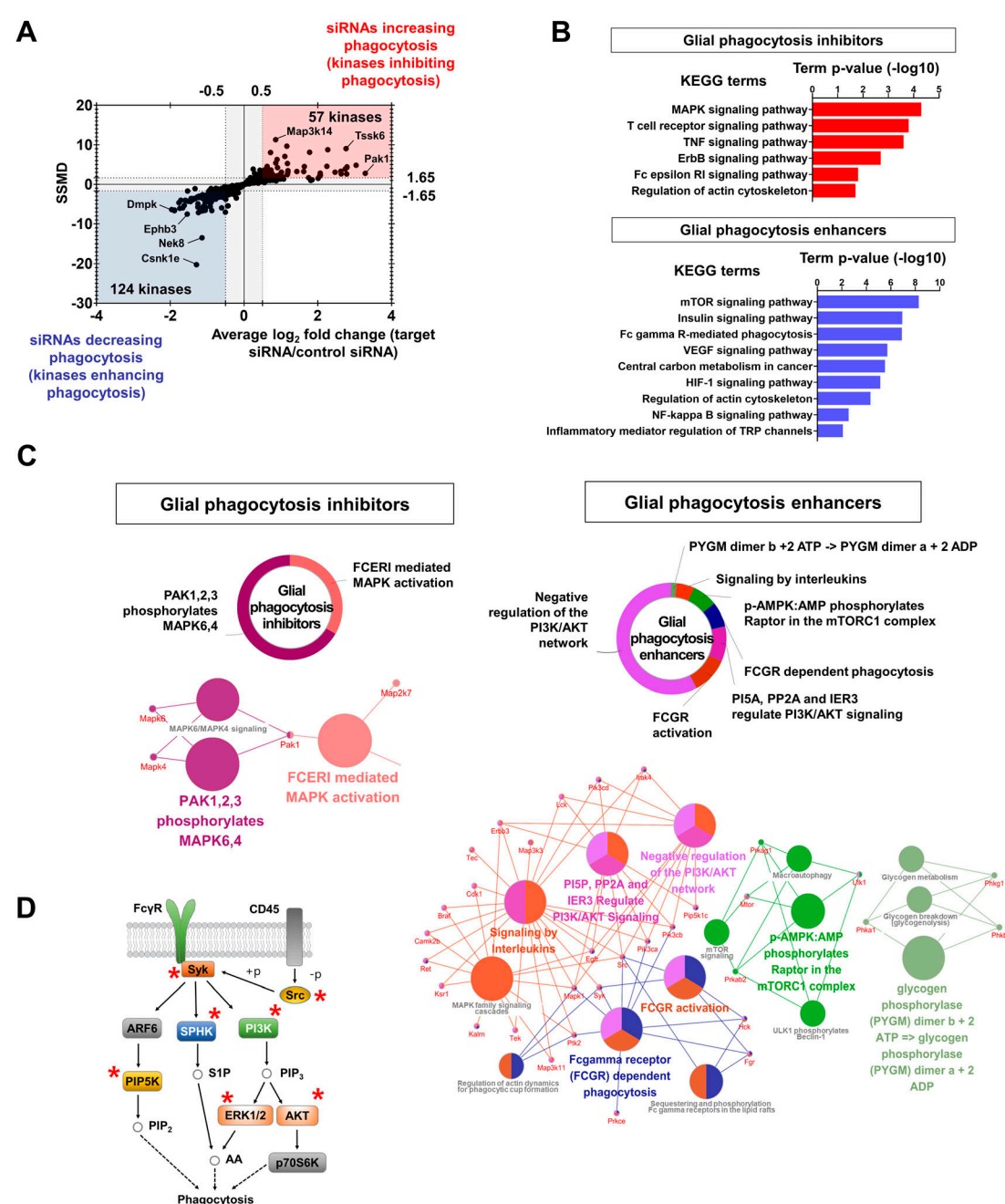

**Figure 5. Kinases regulating glial phagocytosis.**
**(A)** Dual-flashlight plot for strictly standardized mean difference versus fold change (log₂ scale) in the phagocytosis assay. Increased or decreased phenotypes were defined by fold changes ≥0.5 or ≤−0.5, respectively (log₂ scale). Values on the x-axis indicate average fold change of three siRNAs for each target kinase. Final hit kinases were selected by strictly standardized mean difference values ≥1.65 or ≤−1.65. **(B)** Representative KEGG pathway analysis using the DAVID bioinformatics tool. **(C)** Representative functional group network views for Gene Ontology (GO) terms generated using ClueGO. Terms are functionally grouped based on shared genes (κ score) and shown in different colors. **(D)** Simplified "Fcγ receptor-mediated phagocytosis" pathway. Asterisks indicate the kinases associated with the signaling pathway, which were identified in the siRNA screen.

According to KEGG pathway and GO enrichment analyses, the "MAPK signaling pathway," "TCR signaling pathway," "TNF signaling pathway," and "ErbB signaling pathway" were the most significant KEGG pathways enriched for kinases inhibiting phagocytosis; in contrast, "mTOR signaling," "insulin signaling," "Fc γ R-mediated phagocytosis," "VEGF signaling," "central carbon metabolism in cancer," "HIF-1 signaling," "regulation of actin cytoskeleton," and "NF-κ B signaling" were significant pathways for kinases enhancing glial phagocytosis (Fig 5B and Table S11). ClueGO analysis showed that several reactome pathways, including "PAK1,2,3 phosphorylates MAPK6,4," "FCERI mediated MAPK activation," "PI3K/AKT signaling network," "FCGR activation," "FCGR dependent phagocytosis," "AMPK signaling," and "signaling by interleukins" were enriched for phagocytosis-

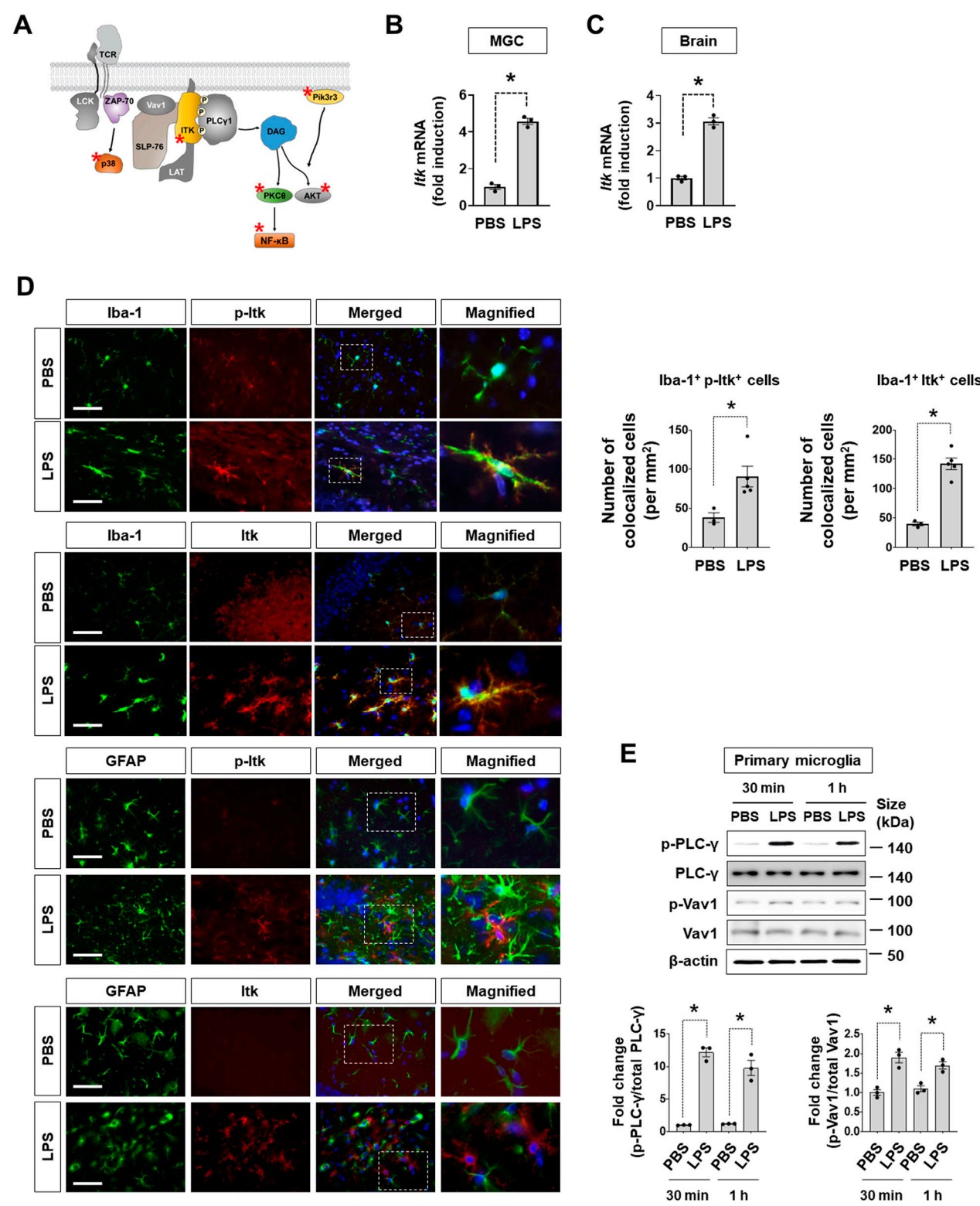

**Figure 6. Itk expression and activity in the brain.**
**(A)** Simplified "T cell receptor signaling" pathway. Asterisks indicate the kinases identified in the siRNA screen. **(B)** *Itk* mRNA levels in MGC stimulated with LPS (1 μg/ml) for 6 h. Data are presented as mean ± SEM (n = 3 replicate wells per group). The assay was repeated at least twice with similar results and the interassay coefficients of variation was <20%, indicating a high level of confidence in the result. **(C)** *Itk* mRNA levels in the inflamed brain 24 h after intraperitoneal LPS injection (5 mg/kg). *Itk* mRNA levels were measured by qRT-PCR in the whole brain. Data are presented as mean ± SEM (n = 3 mice per group). **(D)** Immunofluorescence analysis of phosphorylated Itk (p-Itk) and total-Itk (Itk) in the hippocampus of LPS-induced inflamed brains. Scale bar = 25 μm. Mice were injected with LPS (5 mg/kg) i.p. At 24 h after LPS injection, brains

regulating kinases (Fig 5C). Moreover, several phagocytosis-enhancing kinases identified in our screen were included in canonical Fc γ R-mediated phagocytosis signaling pathways (Fig 5D).

### Role of the identified kinase hits in other aspects of neuroinflammation

In the present study, MGCs were stimulated with the bacterial product LPS. To assess the phenotypes of glia activated by endogenous stimuli, we conducted additional experiments (NO and migration assays) using either high-mobility group box 1 (HMGB1) protein (Pedrazzi et al, 2007) or cell lysates (Eppensteiner et al, 2019) as representative sources of damage-associated molecular patterns (DAMPs). The siRNAs of the 12 kinases used in a previous validation experiment (Fig S3A) were added to MGC cultures, which were then incubated with HMGB1 (20 μg/ml) or cell lysates (1 μg/ml) for measuring NO release and glial migration. As a result, LPS and DAMPs showed similar overall response patterns after kinase knockdown in the MGC culture (Fig S7A).

Our multiplex screen was composed of four different assays for distinct glial phenotypes related to neuroinflammation. However, to further evaluate the role of the identified kinase hits in other aspects of neuroinflammation, we conducted a superoxide assay and ELISA measurement of TNF-α or matrix metalloproteinase (MMP)-9 as representative cytokines or proteases, respectively, using these 12 previously-selected kinase siRNAs (Fig S7B). The results were then compared with those of the NO assay in our screen. The data consistency among the four different assays was as follows: NO assay versus superoxide assay, 83.3%; NO assay versus TNF-α ELISA, 91.7%; and NO assay versus MMP-9 ELISA, 91.7%. Thus, we observed relatively high consistency between the NO assay and the other assays measuring different aspects of neuroinflammation.

### Comparison of glial phenotype screens between MGCs and single cell type culture

To test whether our screening system can be extended to single glial cell types, we compared glial phenotype screens in cultures of a single cell type with that of MGCs after introducing the siRNAs of the 12 representative kinases previously used (Fig S3A). The changes in NO production in MGC culture after the knockdown of kinases were similar to those of microglia (consistency = 50%) rather than astrocytes (consistency = 33.3%). These results suggest that the NO production of MGCs may be more closely related to microglia and their responses than to astrocytes (Fig S8A). The effects of kinase knockdown on MGC migration were also similar to those for microglia (consistency = 75%) and astrocytes (consistency = 91%; Fig S8B). Furthermore, we assessed cell death in cultures of a single

cell type (microglia vs. astrocytes). Then, the top five kinases identified in MGC cytotoxicity assay (~50% cell death) were selected for validation experiments with cultures of a single cell type (Fig S8C). Our data revealed that both microglia and astrocytes were similarly sensitive to cytotoxicity after kinase knockdown.

### Role of microglial interleukin-2-inducible T-cell kinase (ITK) in neuroinflammation

In the NO assay, the "TCR signaling pathway" was enriched in glia-activating kinases (Fig 2B). Indeed, our siRNA screen identified several TCR signaling-associated kinases, such as tyrosine-protein kinase Akt1, Ikbkb, Itk, Pik3r3, MapK11, and Mapk13 as glial activators (Fig 6A). Among them, the knockdown of interleukin-2-inducible T-cell kinase (Itk) produced the most potent effects in the NO assay. Because Itk is a well-known component of TCR signaling and its CNS expression has not been reported, we first investigated whether Itk is expressed in glia and the CNS in vivo. After treatment with LPS for 6 h, *Itk* mRNA expression was increased in MGCs and the inflamed mouse brain 24 h after LPS injection (i.p.; Fig 6B and C). Immunohistochemical analysis identified Itk expression and phosphorylation in microglia, but not astrocytes, in the hippocampal region of the inflamed brain after LPS injection (i.p.; Fig 6D). Furthermore, phosphorylation of PLC-γ and Vav-1, major components of the TCR signaling pathway, was increased in mouse primary microglial cells stimulated with LPS (Fig 6E).

To further investigate the role of Itk in microglial activation, commercially available Itk inhibitors (BMS509744 and GSK2250665A) were applied; these inhibitors significantly reduced LPS-induced NO production in a dose-dependent manner in primary microglia (Fig 7A). The effective concentration was 0.01 μg/ml (6 nM) for BMS509744 and 0.8 μg/ml (1.6 μM) for GSK2250665 in primary microglial cells. Similarly, BMS509744 blocked LPS-induced PLC-γ phosphorylation (Fig 7B), implying that Itk-mediated PLC-γ phosphorylation is involved in microglial activation. To determine the IC50 of the Itk inhibitor, we conducted two different cell-based Itk activity assays (Fig S9A). In the first, Itk inhibition by BMS509744 was quantified in microglial cell lysates using a commercially available Itk kinase activity assay kit. In the second, Itk activity was measured in a Western blot of phospho-PLC-γ (Tyr783), a direct substrate of Itk in microglia. In both assays, the IC50 of BMS509744 in microglial cells was around 100–400 nM, substantially higher than the reported IC50 (~19 nM) in the cell-free system. Accordingly, we speculate that the effective concentrations of the Itk inhibitor may differ between cell-free conditions and microglial cells, supporting the observed relationship between Itk inhibition and its effects on NO production in microglia.

Next, we sought to determine the role of Itk in microglia-mediated neuroinflammation. In the systemic LPS injection-induced neuroinflammation model, co-administration of the Itk

were prepared for immunofluorescence analysis. Glial cells were stained with anti-Iba-1 (microglia) and anti-GFAP (astrocytes) antibodies. A quantification of microglial cells (Iba-1[+]) co-stained with p-Itk (Iba-1[+] p-Itk[+]) or Itk (Iba-1[+] Itk[+]) is also shown (right). Data are presented as mean ± SEM (PBS, n = 3 mice; LPS, n = 5 mice; each data point indicates the average values of six fields of view per mouse). **(E)** Immunoblots of kinases related to the T cell receptor signaling pathway. Primary microglia were treated with LPS (1 μg/ml). Total protein was harvested at the indicated time points and subjected to immunoblot analysis (upper) and quantification (lower). p-PLC-γ, phosphorylated PLC-γ; PLC-γ, total PLC-γ; p-Vav, phosphorylated Vav1; Vav1, total Vav1. Data are presented as mean ± SEM (n = 3 replicate wells per group). **(B, C, D, E)** Unpaired *t* test (B, C, D), One-way ANOVA followed by Tukey's post hoc test (E), *P < 0.05. The assay was repeated at least twice with similar results and the interassay coefficients of variation was <20%, indicating a high level of confidence in the result.
Source data are available for this figure.

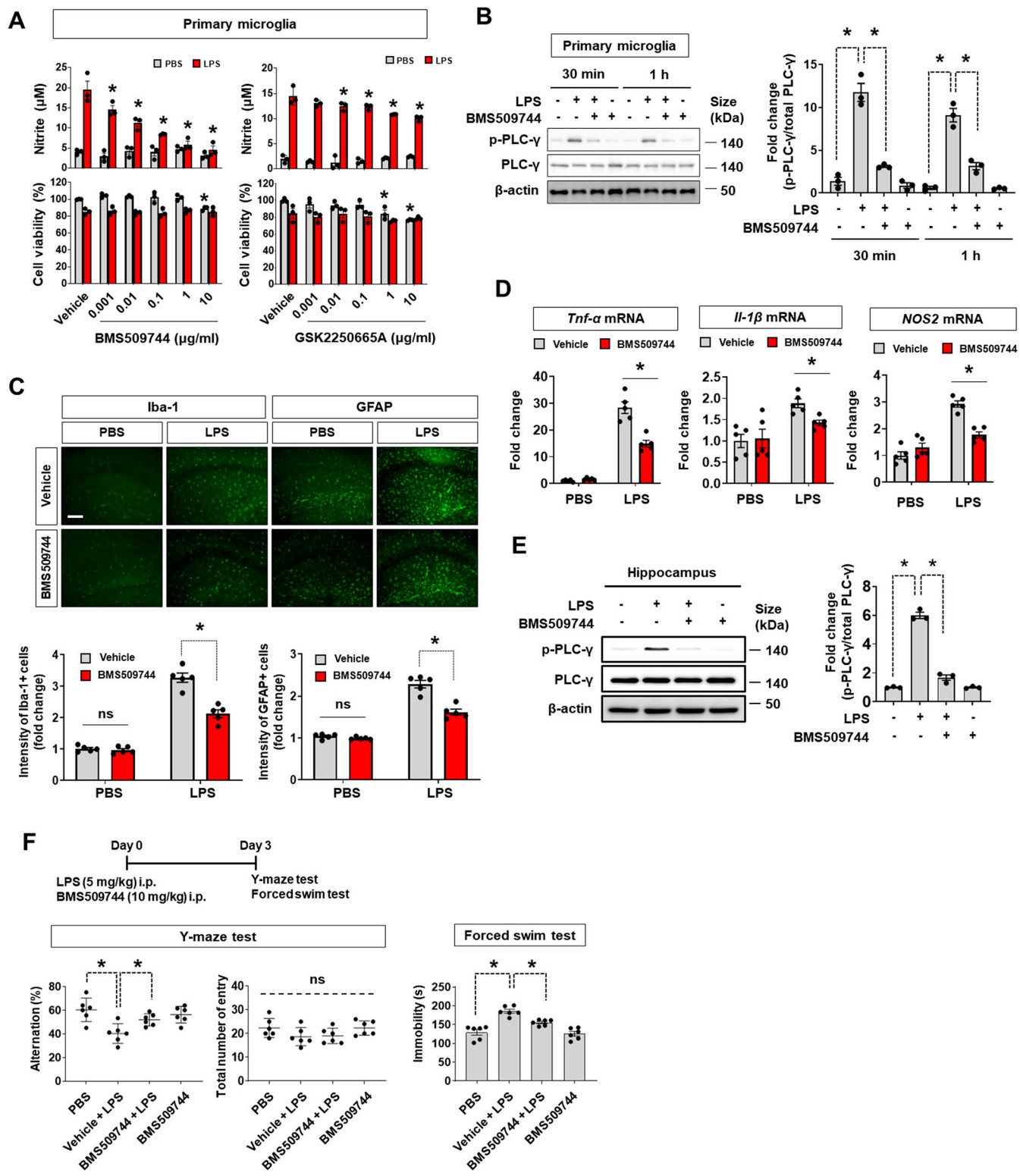

**Figure 7. Inhibition of microglial Itk alleviates LPS-induced neuroinflammation.**
**(A)** Interleukin-2-inducible T-cell kinase (Itk) inhibitors (BMS509744 and GSK2250665A) diminished LPS-induced NO production in primary microglial cells. The cells were stimulated with LPS (1 μg/ml) for 24 h. NO production was measured by nitrite concentration in the cultured media and cell viability was assessed using an MTT assay. Data are presented as mean ± SEM (n = 3 replicate wells per group). **(B)** Immunoblots of PLC-γ (an Itk substrate) after treatment of primary microglia with LPS (1 μg/ml) in the presence or absence of the Itk inhibitor BMS509744 (1 μg/ml). The quantification is shown in the adjacent graph. Data are presented as mean ± SEM (n = 3 replicate wells per group). **(C)** Immunohistochemistry of glial cells in the hippocampal region. Mice were i.p. injected with LPS (5 mg/kg) with or without BMS509744 (10 mg/

inhibitor BMS509744 (10 mg/kg) significantly reduced the activation of microglia and astrocytes. Glial activation characterized by hypertrophy or "gliosis" (Silver & Miller, 2004; Pekny & Nilsson, 2005) was measured based on the immunofluorescence intensity of Iba-1-positive microglia and GFAP-positive astrocytes (Fig 7C). These data suggest that the Itk inhibitor inhibited not only microglial activation but also microglia-mediated astrocyte activation. Moreover, the inhibitor decreased the mRNA expression of the proinflammatory cytokines *Tnf-α*, *Il-1β*, and *Nos-2* in the neuroinflammation model (Fig 7D). LPS-induced PLC-γ phosphorylation in the brain was significantly blocked by the Itk inhibitor (Fig 7E). Because Itk has previously been associated with NF-κB signaling (Wang et al, 2017), Itk inhibition or knockdown may suppress NF-κB-dependent expression of inducible nitric oxide synthase (iNOS) and proinflammatory cytokines in microglia (Fig S9B). Next, we tested whether neuroinflammation-associated cognitive impairment and depressive-like behavior can be relieved by Itk inhibition in mice (Fig 7F). In a Y-maze test, LPS-injected mice showed a significantly lower percentage of spontaneous alternation compared with control mice (PBS and vehicle-injected mice), indicating that the former were cognitively impaired. However, Itk inhibitor injection alleviated this cognitive deficit. Furthermore, in a forced swim-induced behavioral despair test, the Itk inhibitor significantly reduced the immobility of LPS-treated mice. These results suggest that microglial Itk is involved in the behavioral deficits observed under the neuroinflammatory condition.

### Pdk2 modulates astrocyte migration by regulating mitochondrial metabolism: Pdk2 favors glycolysis and inhibits astrocyte migration

Among the kinase hits and associated biological pathways identified in the wound healing assay, we focused on carbon (glucose) metabolism, as it may be potentially involved in the regulation of cell migration (Turner & Adamson, 2011; Rash et al, 2018; Kaushik et al, 2019; Qiao et al, 2019). Hexokinase, pyruvate kinase M, and pyruvate dehydrogenase kinase isoform 2 (Pdk2) were identified in our screen as some of the most potent kinases for deceleration of glial migration (Figs 4B and 8A). All these kinases are closely associated with the glycolytic metabolism. Hexokinase and PKM may indirectly regulate the phosphorylation of other proteins (Roberts & Miyamoto, 2015; Zhang et al, 2019), although these enzymes do not directly phosphorylate proteins as regulatory protein kinases. In particular, Pdk2 has been implicated in neurological diseases (Rahman et al, 2016) and plays a crucial role in macrophage polarization to the M1 phenotype (Min et al, 2019).

For further investigation of the regulatory role of Pdk2 in glial migration, we used a pharmacological inhibitor of Pdk2 (AZD7545) and *Pdk2* gene-deficient glial cells. In the wound healing assay, AZD7545 treatment or *Pdk2* gene KO significantly increased MGC migration after 48 h of LPS-priming (Fig 8B–D and Video 1). As the glucose metabolism has been suggested to play a role in cell migration (Urra et al, 2018), we compared glial cell migration after treatment with a glycolysis inhibitor (2-DG) or oxidative phosphorylation (OXPHOS) inhibitors (FCCP). As shown in Fig 8B–D, the up-regulation of glial migration caused by AZD7545 or *Pdk2* KO was further enhanced by 2-DG but abrogated by oligomycin or FCCP.

Next, we evaluated the comparative movement of glial cells in vivo using a cortical needle injury model. The cortical injury was concurrently generated by local injections of vehicle or AZD7545 into the prefrontal cortex of WT mice. *Pdk2* KO mice were injured by needle insertion alone in the same area of the brain (Fig 8E). Pdk2 expression was mainly observed in reactive astrocytes, as revealed by a high level of GFAP expression, which was used as a standard marker for reactive astrocytes (Liddelow & Barres, 2017) in this model (Fig S10A). The number of glia was immunohistochemically evaluated by counting GFAP- or Iba-1-positive cells in concentric circles from the injection site. BrdU staining was performed to detect proliferative cells. At 24 h after the cortical needle injury, the accumulation of astrocytes around the injury site was significantly increased in AZD7545-injected or *Pdk2* KO mice compared with WT animals (Fig 8E). Moreover, BrdU staining indicated that glial cell migration was not greatly affected by cell proliferation (Fig S10B). Equivalent results were not observed in microglia (Fig S10C). Therefore, Pdk2 inhibition apparently promotes astrocyte migration toward the injury site, which is consistent with the data obtained from cultured glial cells.

We also sought to determine the mechanism underlying the effects of Pdk2 on astrocyte mobility by assessing the mitochondrial metabolic state of astrocytes using a Seahorse XF 24 extracellular flux analyzer (Fig 8F). After 16 h of LPS treatment, the oxygen consumption rate (OCR) was significantly higher in *Pdk2* KO astrocytes than in the WT (Fig 8F). These data imply that *Pdk2* KO astrocytic metabolism is potentially dominated by OXPHOS, with reduced glycolysis-based lactate production. Collectively, these results suggest that Pdk2 modulates astrocyte migration by regulating the mitochondrial metabolism; Pdk2 favors glycolysis and inhibits astrocyte migration.

## Discussion

In this study, we provide an example of an effective method for identifying key kinases regulating the glial phenotypes underlying neuroinflammation. Specifically, we used a novel multiplex screen

kg, i.p.). Astrocytes and microglia in the hippocampus were immunostained with anti-GFAP and anti-Iba-1 antibodies, respectively, at 24 h after LPS treatment. Scale bar = 400 μm. The quantification of glial activation is shown in the adjacent graph. Data are presented as mean ± SEM (n = 5 mice per group; each data point indicates the average value of six fields of view per mouse). **(D)** Expression of proinflammatory genes (*Tnf-α*, *Il-1β*, and *Nos2*) in mouse brains. After LPS injection (i.p., 24 h), hippocampal tissues were subjected to RT–PCR analysis. Data are presented as mean ± SEM (n = 5 mice per group). **(E)** PLC-γ immunoblots in hippocampal tissues at 24 h after LPS injection (5 mg/kg, i.p.) with or without the Itk inhibitor BMS509744 (10 mg/kg, i.p.). The quantification is shown in the adjacent graph. Data are presented as mean ± SEM (n = 3 mice per group). **(F)** Effect of Itk inhibition on LPS-induced behavioral impairments. After 3 d of LPS treatment (5 mg/kg, i.p. injection) with/without the Itk inhibitor BMS509744 (10 mg/kg, i.p.), spatial working memory (Y-maze test) and depression-like behavior (forced swim test) were measured. Data are presented as mean ± SEM (n = 6 mice per group). The Y-maze spontaneous alternation test showed that impaired spatial memory after administration of LPS was reversed by BMS509744 injection (left panel). The forced swim test revealed that the significant increase in immobility observed in LPS-injected mice was alleviated by BMS509744 injection (right panel). **(A, B, C, D, E, F)** Two-way ANOVA followed by Tukey's post hoc test (A), one-way ANOVA followed by Tukey's post hoc test (B, E, F), unpaired *t* test (C, D). *$P < 0.05$. The assays (A, B) were repeated at least twice with similar results and the interassay coefficients of variation was <20%, indicating a high level of confidence in the result. Source data are available for this figure.

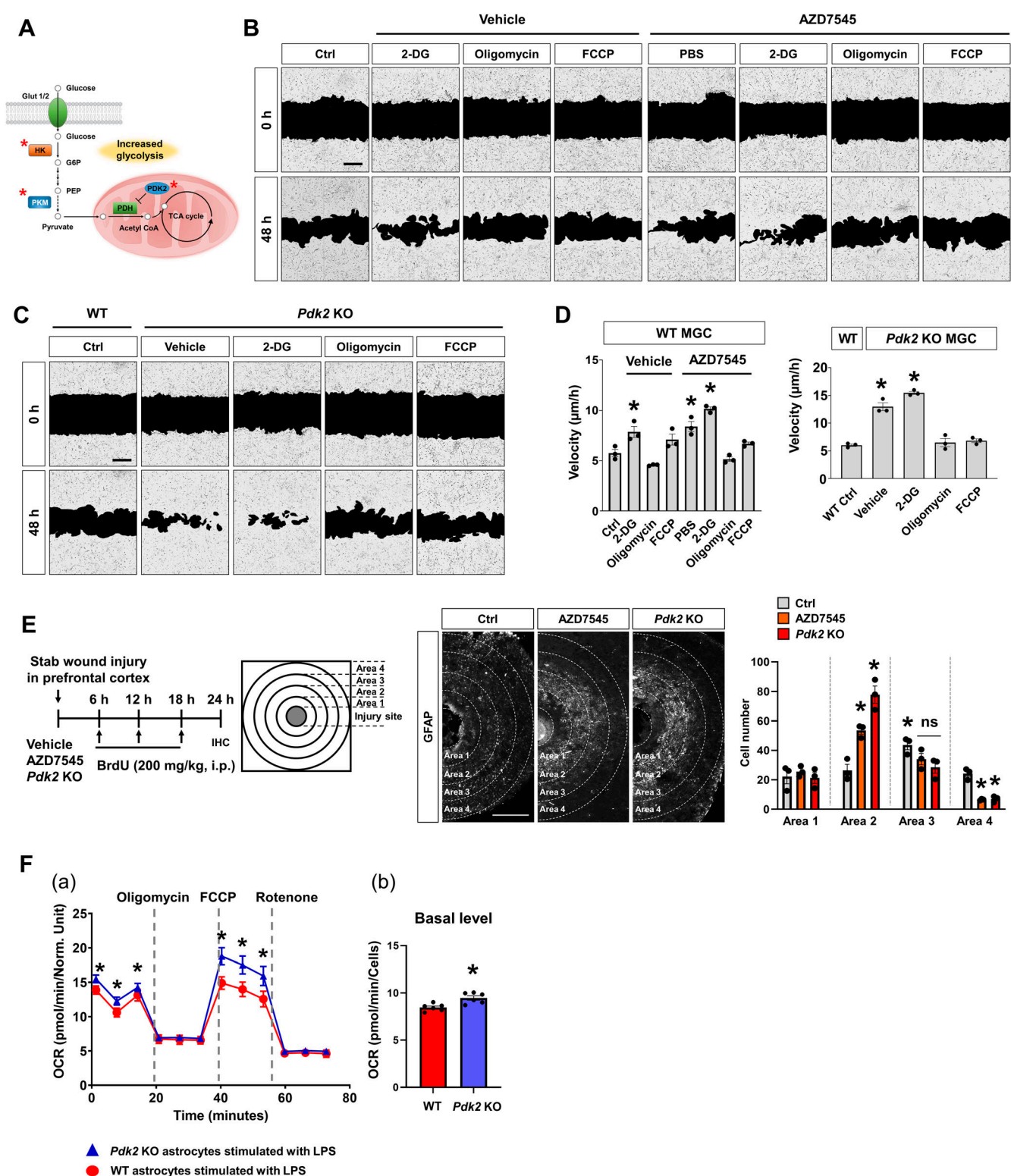

**Figure 8. Role of pyruvate dehydrogenase kinase isoform 2 (Pdk2) in glial cell migration and glycolytic metabolism.**
**(A)** Simplified "Central carbon metabolism in cancer" pathway. Asterisks indicate the kinases associated with specific signaling pathways, identified in the siRNA screen.
**(B)** Representative images from the wound healing assay with mixed glial cells from WT mice. The cells were incubated with a Pdk2 inhibitor (AZD7545, 1 μM), glycolysis inhibitor (2-DG, 2 mM) or an OXPHOS inhibitor (oligomycin or FCCP, 1 μg/ml) for 48 h. Scale bar = 100 μm. **(C)** Representative images from the wound healing assay of mixed glial cells from *Pdk2* KO mice. **(B)** The cells were incubated with the glycolysis inhibitor or OXPHOS inhibitor for 48 h as described in (B). Scale bar = 100 μm. **(B, C, D)** Quantification of migration velocity (μm/h) in (B, C). Data are presented mean ± SEM (n = 3 replicate wells per group). One-way ANOVA followed by Tukey's post hoc test,

in which four different assays were performed in parallel or tandem sequence. Through it, we identified a large number of kinases that positively or negatively regulate glial phenotypes.

We used a systematic and stringent screening approach and identified the essential kinases among the 623 kinases (100%) acting as glial activators (13.8%) or inhibitors (2.6%), key regulators of glial cell death/survival (2.4%), glial migration accelerators (12%) or inhibitors (3.5%), and phagocytosis enhancers (9.1%) or suppressors (19.9%). Validation experiments using pharmacological inhibitors of the representative kinases revealed phenotypic changes similar to those identified in the multiplex kinase siRNA screen. Our multiparametric screening strategy is faster and more efficient than conventional assays that have until now been the workhorses for measuring neuroinflammatory indicators. Conventional assays individually measure single parameters and can be laborious and inefficient in terms of time and cost. The merit of our multiplex screen is its ability to fill the gap between in vitro and in vivo experiments as it runs across multiple stages of in vivo events, such as cell activation, migration, and phagocytosis, which generally occur sequentially in vivo. The multiplex screen could also help alleviate the limitation of data variation stemming from and depending on experimental batch conditions such as cellular population or culture environment. In summary, our multiplex screen enables analyses of several consecutive in vivo events in cultured cells in a time- and cost-efficient manner with minimal experimental variation.

In our screen, the first round of assays (the NO assay) identified crucial regulators of glial cell activation. NO plays a critical role in several physiological and pathological processes in glial cells. It is synthesized mainly by iNOS through TLR4 signaling in response to LPS exposure. Glia-derived NO has significant pathophysiological implications (Calabrese et al, 2007). In the present study, functional network analysis of the glia-inhibiting kinases identified in the NO assay revealed a high enrichment of the phosphatidylinositol signaling pathway. Previous reports demonstrated that LPS triggers a transient activation of PI3-kinase in macrophages (Diaz-Guerra et al, 1999) and microglia (Saponaro et al, 2012); conversely, using specific inhibitors of PI3-kinase, namely, wortmannin or LY294002, results in up-regulation of iNOS expression (Diaz-Guerra et al, 1999). This indicates that PI3-kinase acts as a negative regulator in NO production. Similarly, our screen identified PI3-kinases and related kinases in the same signaling pathway as major inhibitory regulators of glial activation. Nevertheless, both the positive and negative roles of the PI3-kinase/AKT cascade in microglial activation have been previously reported (Calabrese et al, 2007). We also found that significant proinflammatory pathways, such as the

HIF-1, TLR, and chemokine signaling pathways, were enriched KEGG pathways for glial activators (Fig 2B). Moreover, we identified several TLR4 signaling-associated kinases, including Ikbkb, Ikbke, Mapk11, Rps6ka5, and Tbk1, as significant regulators of NO production (Fig S4A). These data imply that our screen successfully identified several kinases that regulate glial activation.

Herein, we also provided insight into Itk signaling pathways in glial cells, particularly microglia. Itk, also known as Emt and Tsk, is a member of the Tec family nonreceptor tyrosine kinases. This kinase is expressed primarily in hematopoietic cells and serves as an important mediator of antigen receptor signaling in lymphocytes (Berg et al, 2005). Itk activation after TCR stimulation has been previously demonstrated (Andreotti et al, 2010). T-cell activation after the binding of antigen to the TCR involves organization of the Vav1–SLP76–Itk complex. This, in turn, activates PLC-γ1, generating IP3 and DAG, which trigger calcium release and PKC activation, respectively. The mobilization of intracellular calcium activates key transcription factors, such as the nuclear factor of activated T lymphocytes, whereas activation of a specific PKC isoform, PKCθ, is associated with distal events leading to NF-κB activation (Paul & Schaefer, 2013). NF-κB plays a critical role in regulating the survival, activation, and differentiation of innate immune cells, such as microglia, and T cells. NF-κB comprises a family of inducible transcription factors regulating a large set of genes involved in various immune and inflammatory response processes. In T cells, the TCR-to-NF-κB pathway involves a series of complex events in which TCR engagement evokes a cytoplasmic cascade of protein–protein interactions and posttranslational modifications, leading to the nuclear translocation of NF-κB. NF-κB then promotes the differentiation of T helper type 1 cells (Oh & Ghosh, 2013). In microglia, NF-κB is a critical mediator of M1-type inflammatory responses triggered by both myeloid differentiation (MYD)88-dependent and toll/IL-1 receptor domain-containing adaptor-inducing IFN-β-dependent TLR signaling pathways (Lin et al, 2012). The MYD88-dependent TLR pathway is crucial for M1 macrophage polarization and is required to induce several inflammatory genes, including those encoding TNF-α, IL-1β, IL-6, IL-12p40, and cyclooxygenase-2. Although the activation and functional mechanisms of NF-κB differ between T cells and microglia, NF-κB is undoubtedly involved in the immune and inflammatory activation of both cell types. In this study, we found that Itk is another common signaling component in both these cell types. Specifically, we demonstrated Itk expression and activation in microglia for the first time. Immunohistochemical analysis revealed that Itk was phosphorylated in activated microglia but not in astrocytes after a systemic LPS challenge. Moreover, PLC-γ (Y783) and Vav1 were

---

*P < 0.05 versus control (Ctrl). The assay was repeated at least twice with similar results and the interassay coefficients of variation was <20%, indicating a high level of confidence in the result. **(E)** Effect of Pdk2 inhibition on astrocyte migration in vivo. Male C57BL/6 mice were intracortically injected with the vehicle or AZD7545 (1 μM). *Pdk2* KO mice were injured by needle stabbing. BrdU was administered i.p. every 6 h after stab-wounding, as indicated. After 24 h, the mice were euthanized. Brain sections were obtained in the transverse plane (tangential to the direction of the needle injury). The quantification of GFAP-positive astrocytes around the injection site in the prefrontal cortex is shown in the right panel. The cell numbers in each concentric circle (radius step size = 500 μm) were analyzed. Data are presented mean ± SEM (n = 3 mice per group). Two-way ANOVA followed by Tukey's post hoc test, *P < 0.05 versus Ctrl in the same area (area 2 or 4); #P < 0.05 versus Ctrl in area 1. **(F)** Effect of Pdk2-deficiency on oxygen consumption rate (OCR) in cultured astrocytes after LPS treatment. Primary astrocyte cultures were prepared from WT or *Pdk2* KO mice and stimulated with LPS (1 μg/ml). A seahorse assay was conducted after 16 h of LPS treatment. Oligomycin was used to inhibit ATP synthase and reduce OCR. FCCP was applied to raise OCR to a maximal value by uncoupling oxygen consumption from ATP production. Rotenone, a complex I inhibitor, was used to completely inhibit the mitochondrial respiration. (a) OCR at different time points. (b) Basal OCR level in the astrocytes of WT or *Pdk2* KO mice. Data are presented mean ± SEM (n = 5 replicate wells per group). Unpaired *t* test, *P < 0.05 versus WT.

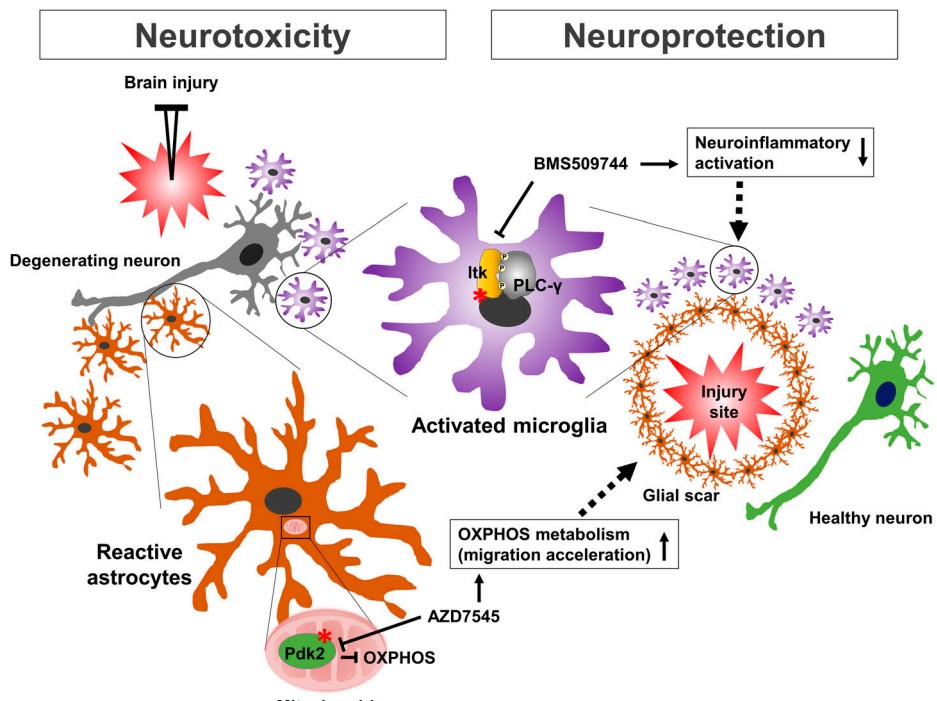

**Figure 9.  Schematic summary showing the relevance of microglial Itk and astrocytic Pdk2 in neuroinflammation and neurotoxicity.**
Brain injury results in the release or leakage of inflammatory molecules from the damaged brain tissue, activating the microglia (through Itk, asterisk). Aberrant microglial activation and phagocytosis can also induce neurotoxicity. This process is followed by astrocytic migration toward the injury site (through Pdk2 inhibition, asterisk). Reactive astrocytes can further cause glial scar formation. The astrocytic scar separates healthy tissue from the damaged tissue, exhibiting neuroprotective effects in brain injury. Thus, Itk inhibition may suppress neurotoxic microglial activation whereas Pdk2 inhibition may promote astrocytic migration and might assist with neurorepair.

phosphorylated in microglial cells after LPS treatment. Furthermore, Itk inhibition mitigated LPS-induced PLC-γ phosphorylation and NO production, mRNA expression of proinflammatory cytokines, and microglial activation. These observations indicate the potential role of the Vav1–SLP76–Itk complex and PLC-γ1 in microglial activation, although the detailed mechanism underlying this observation remains unknown and should therefore be investigated further.

In the multiplexed screen, the wound healing assay showed that knocking down kinases involved in cellular metabolism alters the glial migration phenotype (Fig 4B). Of the kinases identified in the screen, we demonstrated the novel role of Pdk2 in glial migration (Fig 8). Genetic or pharmacological inhibition of Pdk2 resulted in accelerated glial migration, which was associated with a metabolic shift toward OXPHOS (Fig 8F). Pdk2 is one of the various PDK isoforms, which are key regulators of the mitochondrial gatekeeping enzyme pyruvate dehydrogenase (PDH) complex. The PDH complex is composed of three catalytic components, PDH (E1), dihydrolipoamide transacetylase (E2), and dihydrolipoamide dehydrogenase (E3). The basic core of the E1 PDH component is a heterotetramer of two α and two β subunits (α2β2) that catalyzes the first step of pyruvate decarboxylation, converting pyruvate to acetyl-CoA to be used in the tricarboxylic acid cycle and OXPHOS. Pdks inhibit the PDH complex by catalyzing the phosphorylation of serine residues in the E1 α subunit. Whereas Pdk-mediated phosphorylation and subsequent inactivation of the PDH complex result in a metabolic shift toward glycolysis, PDH dephosphorylation by PDH phosphatase induces entry into OXPHOS. Thus, as demonstrated in the current study, either pharmacological Pdk2 inhibition via siRNA or gene KO may promote the up-regulation of PDH activity, thereby inducing a

metabolic shift toward OXPHOS, a process associated with accelerated glial migration.

Recently, the metabolic plasticity of glia has been suggested. Multiple lines of evidence have elucidated the metabolic switch in the neurotoxic activation of microglia and astrocytes. (Ghosh et al, 2018; Morita et al, 2019). Under neuroinflammatory conditions, activated microglia and astrocytes prefer aerobic glycolysis over OXPHOS. Recently, it has been suggested that reestablishing OXPHOS in microglia may improve their migratory activity and their phagocytic potential in Alzheimer's disease (Pan et al, 2019). Cell migration is an energetically expensive process that requires considerable amounts of ATP for cytoskeletal rearrangement. Glial cell migration is crucial to clearing cellular debris through phagocytosis and reestablishing homeostasis in the microenvironment of injured CNS tissue (Saadoun et al, 2005; Morizawa et al, 2017). In line with these reports, we showed that Pdk2 inhibition increases astrocytic migration toward the site of brain injury, indicating a likely role of OXPHOS and ATP in alternative or beneficial astrocyte activation.

Using MGC cultures can be a limitation in screening experiments, as MGCs are not a homogeneous single cell type. Nevertheless, the complexity of this culture may be useful for interpreting in vivo glial responses. Previous studies have used the MGC culture to incorporate complex cell behaviors in vivo into in vitro models and to mimic the key properties of CNS responses (Chen et al, 2013). Thus, MGCs can be used to investigate integrated glial functions and cell–cell communication, given that they mimic the heterogeneous nature of CNS glial cell types, and may therefore help improve current understanding of the mechanistic basis of glial responses (when used in combination with in vivo approaches). Moreover, in vitro culture of single glial cell types can potentially involve contamination with other glial cell types; thus, the robustness of

the MGC culture as an in vitro model is another of its advantages. Although the MGC culture model may have some limitations, evidence suggests that it is a reasonable model for in vitro screening for the discovery of initial candidates, provided that the MGC-based screen must be combined with subsequent validation studies involving single glia cell types and in vivo experiments.

In our multiplexed screen results, it is not clear whether different kinase hits act in the same cell, making it difficult to interpret them as "pathways." Different types of glia, particularly microglia and astrocytes, not only play distinct roles in neuroinflammation but also signal to each other; thus, their contribution to neuroinflammation is non-cell autonomous. The clustered functional pathways in each assay reported in this study potentially provide insight into the overall glial function, including the collective dynamics of a multicellular system (Bich et al, 2019). In addition, the data could indicate the multicellularity of glia, showing cell–cell communication and other non-cell autonomous properties, among others. Interglial crosstalk is crucial for brain development, function, and disease (Jha et al, 2019). Microglia determine the functions of reactive astrocytes, from neuroprotective to neurotoxic (Jha et al, 2018). Conversely, astrocytes regulate microglial phenotypes and functions, such as motility and phagocytosis, through their secreted molecules (Jha et al, 2018). Therefore, the data from the present study improve current understanding on the participation of glial cells in neuroinflammation and related diseases in vivo.

Glial cells show extensive heterogeneity and phenotypic plasticity in neuroinflammation. In addition, diverse glial phenotypes, including activation, migration, proliferation, death/survival, and phagocytosis, all are involved in the regulation of neuroinflammation. In the present study, our multiplexed screen identified a number of kinases that modulate these diverse glial phenotypes. Some kinases uniquely regulate a single glial phenotype, whereas others simultaneously control multiple glial phenotypes (Fig S11 and Table S12). As kinases are the single most important component of intracellular signaling pathways, the glia-regulating kinases identified in this study will improve our understanding of the regulatory signaling mechanisms underlying specific glial phenotypes. Moreover, these kinases could be a valuable source of druggable targets for a glia-based therapy of brain disorders. A search for human diseases associated with these glial kinases in DisGeNET, a database of gene-disease association, revealed a high relevance to common neurological diseases, such as Alzheimer's disease, Parkinson's disease, schizophrenia, and depressive disorders (Fig S12 and Table S13). The disease associations of our kinase hits provide further insight into the role of glial phenotypes and neuroinflammation in neurodegeneration and other common neurological disorders.

Our findings suggest that Itk and Pdk2 are potential kinase targets for therapeutic modulation of the neuroinflammatory phenotypes of glia (Fig 9). Itk inhibition may suppress neurotoxic microglial activation, whereas Pdk2 inhibition may promote astrocytic migration, aiding neurorepair (Renault-Mihara et al, 2011; Chiareli et al, 2021). Combination therapies targeting these two kinases may exhibit enhanced therapeutic effects. Therefore, the present multiplexed glial kinase screen not only provides a therapeutic roadmap for targeting glial kinases to halt neuroinflammation and treat the related CNS diseases but also constitutes a more efficient approach to the cell-based phenotypic screen.

# Materials and Methods

### Reagents

LPS from *Escherichia coli* 0111:B4, oligomycin, carbonyl cyanide-p-trifluoromethoxyphenylhydrazone (FCCP), rotenone, mitomycin C, HMGB1, and BrdU were purchased from Sigma-Aldrich. The following kinase inhibitors were purchased from Tocris: leucine-rich repeat kinase 2-IN-1, BMS509744, GSK650394, and GSK2250665A. IRAK-1/4 inhibitor 1 was purchased from Sigma-Aldrich, SBI-0206965 and GNF-2 were obtained from Selleckchem, P-M2tide was obtained from Enzo Life Sciences, and MlkL inhibitor was obtained from Calbiochem. AZD7545 was kindly provided by Professor In-Kyu Lee at Kyungpook National University.

### Glial cell cultures

Whole brains from 3-d-old C57BL/6 mice were minced and mechanically disrupted using a nylon mesh. The cells obtained were seeded in culture flasks containing DMEM supplemented with 10% heat-inactivated fetal bovine serum, 100 U/ml penicillin, and 100 $\mu$g/ml streptomycin and grown at 37°C in a 5% $CO_2$ atmosphere. The culture medium was changed initially after 5 d and then every 3 d. Cells were used after 14–21 d of culture. To check the cell-specific population, MGC were immunostained with anti-GFAP (mouse IgG, 1:1,000; BD Biosciences), anti-Iba-1 (rabbit IgG, 1:1,000; Wako) or anti-Olig2 (goat IgG, 1:500; R&D systems) antibodies for astrocytes, microglia, and oligodendrocytes, respectively. This was followed by incubation for 2 h at room temperature with the following fluorescence-conjugated secondary antibodies: FITC-conjugated anti-mouse (donkey IgG, 1:500; Jackson Immuno Research Laboratories), Cy3-conjugated anti-goat (donkey IgG, 1:500; Jackson Immuno Research Laboratories), or Cy5-conjugated anti-goat (donkey IgG, 1:500; Jackson Immuno Research Laboratories). DAPI was used for counterstaining (blue). Pure astrocyte cultures were prepared from MGCs by shaking overnight. Pure microglia cultures were obtained from MGCs by mild trypsinization (Saura et al, 2003). The BV-2 immortalized mouse microglial cell line was maintained in DMEM supplemented with 10% FBS, 100 U/ml penicillin, and 100 $\mu$g/ml streptomycin.

### Kinase siRNA libraries

The siRNA libraries of 623 mouse kinases (MISSION siRNA Mouse Kinase Panel, SI42050) containing three different sequences of siRNAs targeting each kinase gene were purchased from Sigma-Aldrich. The siRNAs were supplied in 96-well plates, diluted to 10 $\mu$M working stocks upon arrival, according to the manufacturer's instructions, and stored at −20°C until use.

## Evaluation of siRNA transfection efficiency

The control siRNA of GAPDH (Life Technologies) was labeled with Cy3 according to the manufacturer's instructions. MGCs were transfected with *Silencer* Cy3-labeled GAPDH siRNA (20 μM) using Lipofectamine RNAiMAX reagent (Life Technologies) overnight. After 48 h, MGCs were immunostained with anti-GFAP (rabbit IgG, 1:1,000; Dako) or anti-Iba-1 (goat IgG, 1:1,000; Wako) antibodies. This was followed by incubation for 1 h at room temperature with the following fluorescence-conjugated secondary antibodies: FITC-conjugated anti-rabbit (donkey IgG, 1:500; Jackson Immuno Research Laboratories) and FITC-conjugated anti-goat (donkey IgG, 1: 500; Jackson Immuno Research Laboratories). DAPI was used for counterstaining (blue).

## High-throughput siRNA screen in a multiplexed format

Before the high-throughput screen, the concentration of siRNAs, transfection reagents, and transfection efficiency was optimized for MGCs. For the screen, MGCs at 14 d in vitro were seeded (2,500 cells per well) into each well of 96-well flat, clear-bottomed and black-walled, polystyrene-treated tissue-culture microplates (Corning). After 48 h, the cells were transfected with siRNAs using Lipofectamine RNAiMAX reagent (Invitrogen), according to the manufacturer's instructions. Afterwards, the following four assays were performed in sequence (Fig 1).

### NO assay
The $NO_2^-$ concentration in culture media was measured to assess NO production in MGCs using Griess reagent. For each sample, 50 μl aliquots were mixed with 50 μl of the Griess reagent (1% sulfanilamide/0.1% naphthylethylene diamine dihydrochloride/2% phosphoric acid) in a 96-well plate. The absorbance at 550 nm was then measured on a microplate reader (SpectraMax M5; Molecular Devices). $NaNO_2$ was used as the standard to calculate $NO_2$ concentrations.

### Cytotoxicity assay
Cytotoxicity was evaluated by measuring the amount of released LDH using the CytoTox 96 Non-Radioactive Cytotoxicity Assay Kit (Promega), according to the manufacturer's instructions. Culture media (50 μl) from the MGCs were incubated with an LDH substrate solution (50 μl) for 20 min in a dark room. After adding the stop solution, the absorbance was measured at 490 nm using a microplate reader (SpectraMax M5; Molecular Devices). The data were calculated using the following formula: % cytotoxicity = 100 × (experimental/maximum LDH release), according to the manufacturer's instructions. Lysis solution (0.8% Triton X-100) was used to generate the maximum LDH release with 100% cell death control.

### Wound healing assay
The in vitro wound healing assay was performed using the IncuCyte ZOOM Live-Cell Imaging system (Essen Bioscience). MGCs were treated with mitomycin C (5 μg/ml) for 2 h before performing a wound to inhibit cell proliferation. Wounds were made with an IncuCyte WoundMaker and plates were automatically analyzed for wound closure using the IncuCyte ZOOM Live-Cell Imaging system.

Real-time images were acquired every 2–3 h for 48 h. Cell confluence was quantified using time-lapse curves generated by the IncuCyte ZOOM software.

### Phagocytosis assay
The phagocytosis assay was performed according to the manufacturer's instructions using pH-sensitive pHrodo Red Zymosan BioParticles (Life Technologies). The pHrodo Red conjugates do not fluoresce outside the cell at neutral pH but do fluoresce at acidic pH values such as those in phagosomes; this enables an accurate measurement of phagocytosis. Briefly, MGCs were incubated with Zymosan conjugate particles (10 μg/ml) diluted in a live-cell imaging solution (Thermo Fisher Scientific) for 2 h in the dark. Nuclei were counterstained using Hoechst (1:1,000; Life Technologies). After incubation, the cells were thoroughly washed and fixed with 2% paraformaldehyde for 10 min at room temperature. Cell fluorescence was then measured with a microplate reader (SpectraMax M5; Molecular Devices).

## Quality control, hit selection, and pathway enrichment analysis

For quality control purposes, the data quality of individual plates was controlled with the SSMD method, and applied to each positive control within a plate (Birmingham et al, 2009). The criteria for quality control were fixed according to the SSMD definition for a "moderate effect." The SSMD method, combined with FC values, was used to rank genes in each screen and performed in triplicate. SSMD cutoffs <−1.65 and >1.65 with FC values of 0.7fold and 1.5fold were used to define inhibitors and enhancers, respectively. The reproducibility of all assays was confirmed by duplicate trials and Pearson's correlation. Pathway enrichment analyses based on GO (Immune systems, Reactome pathway, and Reactome) or KEGG pathways were performed using the DAVID bioinformatics resource (Huang da et al, 2008) and the ClueGO plug-in for Cytoscape software (Bindea et al, 2009).

## Mice and animal care procedures

Male C57BL/6 mice (8 wk old) were obtained from Samtaco. Male pyruvate dehydrogenase kinase isoform 2 (*Pdk2*) KO mice aged 8–10 wk were used after generation as previously described (Go et al, 2016). Age-matched WT mice were produced from C57BL/6J mice (The Jackson Laboratory), used to stabilize the genetic background of the *Pdk2* KO. Genotypes were confirmed by PCR of the genomic DNA as previously described (Go et al, 2016). Animals were housed under a 12-h light/dark cycle (lights on from 07:00–19:00) at a constant ambient temperature of 23°C ± 2°C with food and water ad libitum. Each individual animal was used for a single experimental purpose. All animal experiments were performed in accordance with the animal protocols and guidelines approved by the Animal Care Committee at Kyungpook National University (No. KNU 2018-0084).

## LPS injection and stab-wound injury model

A systemic LPS injection was performed to evoke neuro-inflammation in mice, as previously described (Jo et al, 2017). Mice received an i.p. injection of vehicle or LPS (5 mg/kg). Animals in the

vehicle control group received the same volume of saline solution. Animals were killed 24 h after injection under deep ether-induced anesthesia. For the stab-wound injury model, mice were anesthetized using inhaled isoflurane (3%) and placed in a stereotaxic device. A stab-wound injury was performed using a needle (30 G) injection. Pdk2 inhibitor (AZD7545) or vehicle was then stereotactically injected (flow rate: 0.1 $\mu$l/min) at a volume of 0.5 $\mu$l into the prefrontal cortical area (anteroposterior: –2.5 mm; mediolateral: 1.5 mm; dorsoventral: –1.0 mm) through a small burr hole. In *Pdk2* KO mice, the injury was generated by inserting the needle only. The skin was sutured after mounting the burr hole using sterile bone wax (Ethicon). To assess glial cell proliferation, BrdU (200 mg/kg) was injected (i.p.) three times every 6 h after surgery. The mice were killed at 24 h after inflicting the stab-wound injury.

## Behavioral tests

For the Y-maze test, the Y-maze comprised of a horizontal maze with three arms (length: 40 cm; width: 3 cm; and wall height: 12 cm). Tested animals were initially placed in the center of the maze and the order (e.g., ABCCAB) and number of arm entries were manually recorded over a period of 7 min for each animal. Voluntary shifts were defined as trials with entries into all three arms in sequence (i.e., ABC, CAB, or BCA, but not BAB). The maze was thoroughly cleaned with water after each test to remove the residual animal odor. The ratio of alternatives was calculated according to the following equation: % alternation = ([number of alternations]/[total arm entries]) × 100. The total number of arm entries was used as an indicator of locomotor activity. All recordings and calculations were automatically obtained by SMART video tracking software version 3.0 (Harvard Apparatus).

For a forced swim test, mice were placed individually in a vertical acrylic cylinder (height: 60 cm; diameter: 20 cm) lled with tap water (26°C) to a depth that did not allow the mice to touch the bottom with their hind paws (20 cm). The animals were removed from the water after 6 min and dried quickly before being returned to their cages. Each session was video-recorded and analyzed blindly. The three following behaviors were considered according to Porsolt's criteria (Can et al, 2012): (i) immobility: mice were considered immobile when they oated passively, making only small movements to keep their nose above the surface; (ii) climbing (or thrashing): defined as upward-directed movements with the forepaws, in and out of the water, and/or along the side of the swim chamber; and (iii) swimming: including active movements (usually horizontal) more than necessary to merely maintain their head above the water. Diving and face-shaking behaviors were not scored. The time spent immobile was measured as well.

## Immunohistochemistry and image analysis

### Double-staining immunohistochemistry

For the immunofluorescence analysis of animal tissues, frozen brain sections (20 $\mu$m) were permeabilized in 0.1% Triton X-100 and blocked using 1% bovine serum albumin and 5% normal donkey serum for 1 h at room temperature. Brain sections were incubated with the following primary antibodies at 4°C overnight: anti-GFAP (rabbit IgG, 1:500; Dako), anti-Iba-1 (goat IgG, 1:500; Wako), anti-

phosphorylated Itk (mouse IgG, 1:200; Thermo Fisher Scientific), anti-Itk (mouse IgG, 1:200; Santa Cruz Biotechnology) or anti-Pdk2 (rabbit IgG, 1:100; Abcepta). This was followed by a 2-h incubation at room temperature with the following fluorescence-conjugated secondary antibodies: FITC-conjugated anti-rabbit (donkey IgG, 1:500; Jackson Immuno Research Laboratories), FITC-conjugated anti-goat (donkey IgG, 1:500; Jackson Immuno Research Laboratories) or Cy3-conjugated anti-mouse (donkey IgG, 1:500; Jackson Immuno Research Laboratories). Finally, the sections were mounted and counterstained using DAPI-containing gelatin.

### Immunohistochemistry of astrocytes and microglia

Brain tissues were processed and sections subjected to immunohistochemical analysis using anti-GFAP (rabbit IgG, 1:500; Dako), anti-Iba-1 (rabbit IgG, 1:500; Wako) or anti-BrdU antibodies (rat IgG, 1:200; Bio-Rad) overnight. This was followed by incubation for 2 h at room temperature with the following fluorescence-conjugated secondary antibodies: FITC-conjugated anti-rabbit (1:200; Jackson Immuno Research Laboratories) and Cy3-conjugated anti-rat (1:200; Jackson Immuno Research Laboratories). Data acquisition and immunohistological intensity measurements were performed using ImageJ, as previously described (Jo et al, 2017). In brief, tiled images of each section were captured using a CCD color video camera (Ximea). An image composite was then constructed for each section using Adobe Photoshop version CS3. The images were binary threshold at 50% of the background level and the particles were then converted to a subthreshold image. Areas <300 pixels and >5 pixels were considered GFAP- or Iba-1-positive cells. The quantification of cells around the needle injection site was performed using an adapted version of Sholl analysis, as previously described, with slight modifications (Jo et al, 2017). Briefly, the number of cells was counted in concentric circles starting from the center of the injury site with a radius step size of 500 $\mu$m. The final radius was set where the cell density reached the normal cell distribution. Proliferating cells were identified by merging GFAP- or Iba-1 staining with BrdU staining.

## Real-time PCR

Real-time PCR was performed using a One-Step SYBR PrimeScript RT–PCR Kit (Perfect Real-Time; Takara Bio), followed by detection using an ABI Prism 7000 Sequence Detection System (Applied Biosystems). Normalization was performed using two internal controls, ribosomal protein lateral stalk subunit P0 (Rplp0) and tubulin $\alpha$ 1A (Tub1a), and a model-based variance and stability calculation (Vandesompele et al, 2002; Andersen et al, 2004). The nucleotide sequences of the primers were based on published cDNA sequences (Table S14).

## Immunoblotting analysis

Cells were lysed in ice-cold RIPA lysis buffer (Thermo Fisher Scientific) and the protein concentration in cell lysates was determined using a Bradford protein assay kit (Bio-Rad). An equal amount of protein (30 $\mu$g per sample) was separated using 12% SDS–PAGE and transferred to polyvinylidene fluoride filter membranes (GE Healthcare). The membranes were blocked using 5%

skim milk and incubated sequentially with the following primary antibodies: anti-phosphorylated-Vav1 (Y174) antibody (mouse IgG, 1: 1,000; Santa Cruz Biotechnology), anti-Vav1 antibody (rabbit IgG, 1: 1,000; Cell Signaling Technology), anti-PLC-γ (Y783) antibody (rabbit IgG, 1:1,000; Cell Signaling Technology); anti-PLC-γ antibody (rabbit IgG, 1:1,000; Cell Signaling Technology), anti-Epha2 (rabbit IgG, 1: 1,000; Cell Signaling Technology), anti-Akt2 (rabbit IgG, 1:1,000; Cell Signaling Technology), anti-Itk (mouse IgG, 1:1,000; Santa Cruz Biotechnology), anti-Pdk2 (rabbit IgG, 1:1,000; Abcepta), anti-Irak4 (rabbit IgG, 1:1,000; Cell Signaling Technology), anti-Mlkl (mouse IgG, 1:1,000; Cell Signaling Technology), and anti-β-actin antibody (mouse IgG, 1: 2,000; Thermo Fisher Scientific). This was followed by a 2-h incubation at room temperature with horseradish peroxidase-conjugated secondary antibodies: anti-rabbit IgG antibody (1:2,000; Cell Signaling Technology) or anti-mouse IgG antibody (1:2,000; Cell Signaling Technology). Immunoblots were developed using a SuperSignal WestPico chemiluminescent substrate (Thermo Fisher Scientific).

### Assessment of extracellular metabolic flux

An XF24 Extracellular Flux Analyzer (Seahorse Bioscience Inc.) was used to determine the OCR, as previously described. Briefly, astrocytes from WT and *Pdk2* KO mice were plated at a density of 40,000 cells per well with LPS in a Seahorse XF24 plate and cultured for 16 h. 1 h before the assay, the medium was exchanged for Seahorse XF basal DMEM containing 1.5 mM sodium pyruvate, 1 mM glutamine, and 25 mM glucose. Rotenone or 2-DG, FCCP, and oligomycin were diluted in XF24 medium with LPS and loaded into the accompanying cartridge to achieve final concentrations of 2, 1, and 1 μg/ml, respectively. Reagent injections into the medium occurred at the time points specified before the analysis start point; subsequently, OCR was monitored. Each cycle was set to mix for 3 min, delay for 2 min, and measure for 3 min. Total protein was extracted from the cells immediately after OCR readings, and results were normalized to the total protein concentration.

### Superoxide assay

Transfected MGC cultures were stimulated with LPS (1 μg/ml) for 30 min and incubated with 10 μM dihydroethidium (Molecular Probes). The cells were then washed with ice-cold PBS and examined with a microreader (excitation: 534 nm; emission: 580 nm) to evaluate superoxide production.

### ELISA

The transfected MGC cells were stimulated with LPS (1 μg/ml) for 48 h and then culture media were harvested. Levels of TNF-α or MMP-9 protein were measured using a mouse TNF-α DuoSet ELISA Kit (R&D Systems) or mouse Total MMP-9 Quantikine ELISA Kit (R&D Systems).

### Itk activity assay

#### Assay 1

BV-2 microglial cells were incubated with LPS (100 ng/ml) and a serially diluted Itk inhibitor (BMS509744). Cell lysates were then harvested in the presence of protease and phosphatase inhibitors. After determination of the optimal amounts of cell lysate (containing Itk and other proteins) necessary to measure Itk activity based on the signal-to-background ratio, the substrate (poly $E_4Y_1$)/ ATP mix was added to the cell lysate and incubated for 1 h. After depleting the remaining ATP, Itk activity was visualized by a kinase detection solution.

#### Assay 2

The cell lysate was subjected to Western blotting for detecting phosphorylated-PLC-γ (Tyr783) (rabbit IgG, 1:1,000; Cell Signaling Technology), a direct substrate of Itk. The IC50 of BMS509744 was calculated based on the inhibition of PLC-γ phosphorylation.

### Quantification and statistical analysis

Data are presented as means ± SEM or ± SD. Data were compared using an unpaired two-tailed $t$ test or ordinary one-way ANOVA followed by Tukey's post hoc test. All statistical analyses were performed using Prism software version 8.0 (GraphPad Software). $P$-values <0.05 were considered statistically significant.

# Supplementary Information

# Acknowledgements

This work was supported by a grant from the Basic Science Research Program through the National Research Foundation (NRF), which is funded by the Korean Government (Ministry of Science and ICT; 2017R1A5A2015391 and 2020M3E5D9079764). DH Park was supported by the Basic Science Research Program of the National Research Foundation of Korea (NRF), funded by the Korean government (Ministry of Science and ICT) (2019R1A2C1084371).

## Author Contributions

J-H Kim: conceptualization, data curation, formal analysis, validation, investigation, visualization, methodology, and writing—original draft.

J Han: data curation, formal analysis, validation, investigation, visualization, and methodology.

R Afridi: data curation, formal analysis, validation, investigation, visualization, methodology, and writing—review and editing.

J-H Kim: data curation, formal analysis, validation, investigation, visualization, and methodology.

MH Rahman: data curation, formal analysis, validation, investigation, visualization, and methodology.

DH Park: data curation, formal analysis, validation, investigation, visualization, and methodology.

WS Lee: data curation, formal analysis, validation, investigation, visualization, and methodology.

GJ Song: data curation, formal analysis, validation, investigation, visualization, and methodology.

K Suk: conceptualization, data curation, supervision, funding acquisition, project administration, and writing—original draft, review, and editing.

## Conflict of Interest Statement

The authors declare that they have no conflict of interest.

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
