## [Reviewer comments · Life Science Alliance]

Life Science Alliance

A multiplexed siRNA screen identifies key kinase signaling networks of brain glia

Jong-Heon Kim, Jin Han, Afridi Ruqayya, Jae-Hong Kim, Md Habibur Rahman, Dong Ho Park, Won Suk Lee, Gyun Jee Song, and Kyoungho Suk

DOI: <https://doi.org/10.26508/lsa.202201605>

Corresponding author(s): Kyoungho Suk, Kyungpook National University School of Medicine

Review Timeline:

Submission Date:	2022-07-13
Editorial Decision:	2022-09-06
Revision Received:	2023-01-29
Editorial Decision:	2023-02-17
Revision Received:	2023-02-22
Accepted:	2023-02-22

Transaction Report:

September 6, 2022

Re: Life Science Alliance manuscript #LSA-2022-01605-T

Prof. Kyoungso Suk
Kyungpook National University School of Medicine
Department of Pharmacology
680 Gukchaebosang Street, Joong-gu
Daegu 41944
KOREA, REPUBLIC OF

Dear Dr. Suk,

Thank you for submitting your manuscript entitled "A multiplexed siRNA screen identifies key kinase signaling networks of brain glia" to Life Science Alliance. The manuscript was assessed by expert reviewers, whose comments are appended to this letter. We invite you to submit a revised manuscript addressing the Reviewer comments.

Thank you for this interesting contribution to Life Science Alliance. We are looking forward to receiving your revised manuscript.

Sincerely,

B. MANUSCRIPT ORGANIZATION AND FORMATTING:

Reviewer #1 (Comments to the Authors (Required)):

Review of "A Multiplexed siRNA screen identified key kinase signaling networks of brain glia"

In this manuscript titled "A Multiplexed siRNA screen identified key kinase signaling networks of brain glia" the authors are interrogating intracellular signaling pathways involved in glial cell activation, an understudied area. In order to do this, the researchers developed a multiplexed kinome wide siRNA screen to identify kinases that regulate inflammatory LPS induced phenotypes of glial populations. Authors used primary mixed glial cultures from embryonic mice, and four different phenotypic assays (Nitric oxide, cell viability/cytotoxicity, migration/wound healing, and phagocytosis/zymosan uptake). Authors also investigated role of ITK and PDK2 in microglial NO production/activation and astrocyte migration respectively in vivo in LPS and stab injury mouse models.

The work has important implications for our understanding of kinases regulating glial phenotypes in reactive states. However, several issues need to be addressed prior to publication:

Major issues

1. The work flow of multiplex siRNA screens should be described in more details. For instance based on Fig 1 authors tested 623 kinases using triplicate siRNA for each, with LPS or PBS treatment, and with all measurements performed in triplicates -to do this authors needed min of 117 96- well plates for one repetition. And to perform four different assays on the 117 wells authors needed total of 350 plates (two plates for supernatant for NO and viability, and original plates for migration and phagocytosis). This is a huge number of samples to process simultaneously and it will be worthy explaining how was this practically done as this could influence results. For example were all 623 kinases tested at the same time for simultaneous processing of 116 96 well plates or was this done over several independent experiments? How many litters of pups were used for all of the experiments? How many plates total were used? What quality checks were being run to ensure that conditions were the same across different plates/culture preparations?
2. Authors should consider switching what is being used for main figures and supplementary figures. While the pathway analysis is interesting, the specific kinases and confirmation of those results would make the paper more easily understandable and more impactful. Therefore showing the actual results of assays from the siRNA screen for the top kinases is highly recommended for Figs 2-5.
3. Authors use less strict statistical analysis for cytotoxicity presented in Fig 3 than the other 3 experiments. Authors need to use consistent method of analysis for all four phenotypes.

Minor:

4. IHC experiments with cultured microglia - it looks like there are very few microglia in images taken of cultured glial cells. Numbers wise, there are only ~20% microglia and ~70% astrocyte populations in these cultures. Is this consistent across different MGC preparations and different untreated/treated wells?
5. Sup. Fig 2 - Western blot confirms the down regulation of specific kinases with the siRNAs in PBS condition. Authors should include LPS treatment and an LPS plus siRNA condition to determine whether LPS affects level of mRNA/protein reduction?
6. Sup. Fig. 3-6 - Each of these supplemental figures has a table with information on whether a specific kinase is either up or down regulated. It would be helpful to make these table more quantitative with FC and or P value. That would make the information more impactful and valuable. Moreover we strongly suggest data to be presented consistently as FC (i.e. Figure S3. please show as FC instead of nitrite concentration) to be comparable with siRNA data.
7. In Sup Fig 3 authors should use two way ANOVA to determine effect of LPS and kinase inhibitors.
8. When calculating Hedges effect size what n was used? For instance each kinase had triplicate siRNA -were all three siRNA was used? Did all three show same results? Please show all data points in histograms (a dot for each biological or technical replicate).
9. Authors state in results on NO production "Overall, we identified 16 kinases that inhibited glial activation (glial inhibitors) and 86 kinases that enhanced glial activation (glial activators)." Indicating that most identified kinases were glial activators/NO-increasing. Later in the same paragraph authors state "Due to the limited number of NO increasing kinases identified in the screen (15.7%), the selected hits were coincidentally either NO-decreasing kinases or kinases without significant effect", suggesting that most were glial inhibitors/NO-decreasing. Could authors please clarify these contradicting sentences?
10. For survival data presented in Sup Fig 3 could authors create table similar to NO (comparing effects of siRNA screen and

inhibitors)?

11. Could authors please clarify why in Fig 6E there is 2 fold increase in PLC phosphorylation 1hr vs 30 min in PBS? What causes this increase in PLC phosphorylation in PBS?
12. Fig 7 should show the effects of ITK inhibition on PLC Phosphorylation in vivo.
13. Could authors please explain the differences in the extent of LPS induced PLC phosphorylation in BV-2 cells (i.e. in Fig 6E it is 2 fold after 1 hr., and in Fig 7B it is 20 fold after 1 hr). This is especially confusing as figure legends states". The assay was repeated at least twice with similar results and the inter-assay coefficients of variation was < 20%, indicating a high level of confidence in the result". Yet these results indicate 10 fold difference in inter-assays.
14. Could authors discuss why in the stab injury microglia do not migrate to the injury site (Figure S8C)?
15. Sup. Fig. 9 and 10 - These figures would be far more impactful if we knew what kinases fell into each cluster and each category. Having a table of that information along with the Venn diagrams and pie charts would make the importance of these figures much more clear.
16. Sup. Fig 11 - this figure should be in the main body of the paper. This is the main conclusion the paper is trying to make, it shouldn't be relegated to the supplemental.

Reviewer #2 (Comments to the Authors (Required)):

The MS by Kim et al presents a novel screen for kinases which are involved in neuroinflammatory processes in glial cells. They use mixed glial cell cultures, which in their hands mainly contain astrocytes and microglia and knock down a total of 623 kinases using siRNA. These cultures are incubated with LPS to mimic an inflammatory event. Four different parameters are assessed in the following which correlate with different aspects of neuroinflammation: 1) NO production via detection of nitrite; 2) cell death via release of LDH; 3) glial migration using a wound healing assay; 4) phagocytosis using labelled zymosan as substrate. The authors identify numerous kinases, which are involved in the different processes. The relevance of few selected kinases is studied in cell culture as well as in vivo experiments using pharmacology and genetic approaches.

Two kinases are studied in more detail:

Interleukin-2-inducible T-cell kinase (ITK) is shown to be involved in neuroinflammation as its KD affects the NO production, which is substantiated by additional experiments. PDK2 is shown to modulate astrocyte metabolism, and astrocyte migration. The authors show that PDK2 KO leads to increased oxygen consumption which is interpreted as a switch from glycolysis to Oxphos. This notion could be substantiated rather easily by measuring glycolytic rate using the same system (Seahorse), which then should show a reduced glycolytic rate. However, a causal connection of the change metabolism and reduced migration (as suggested by the authors) is not shown by experimental data. It might be independent processes regulated by the same kinase(s). This should be more carefully stated and discussed, or -alternatively- the causal relation should be shown experimentally.

Additional issues which should be addressed:

- 1) The author state that their cultures contain astrocytes (71.5%) and microglia (27.5%). They should at least comment that e.g. oligodendrocytes are lacking, which have been shown to be involved in neuroinflammation as well. Furthermore, no information is given on how many cultures were quantified and how large the variability of these cultures is. Therefore, more detailed statistical information needs to be given here. This also applies to the statement on transfection efficiency: "51.1% and 72.7% transfection rates".
- 2) In general, it is hard to understand why in some cases a subset of kinases is selected. E.g. in Fig S6, 12 kinases are studied in panels A, B, but only 5 in panel C. A rationale and the criteria for these selections (not only for Fig. S6 but throughout the paper) would be helpful.
- 3) Experiments shown in Fig. 7C and corresponding text: The number of microglia increases by a factor of 3, the number of astrocytes by a factor of 2 when exposed to LPS. The authors state that this is due to "activation". However, such an increase in cell numbers most likely includes massive proliferation, as microglia are Iba1+ even in the quiescent state and -in the hippocampus, which is studied here- also most astrocytes are GFAP+ under basal conditions.
- 4) Hexokinase, pyruvate kinase M, and pyruvate dehydrogenase kinase isoform 2 are identified as most potent kinases in the deceleration of glial migration. The authors state that these are "associated with glycolytic metabolism". This is essentially true, but it might be helpful to mention that hexokinase and pyruvate kinase M are actually enzymes within the glycolytic pathway which do not phosphorylate proteins as regulatory kinases.
- 5) When using the glycolysis inhibitor 2DG, the authors need to indicate if there are any other substrates in addition to glucose within the medium, and -if so- which ones. If 2DG is used in the presence of glucose only, also mitochondria are depleted from their substrates as pyruvate is not produced from glycolysis anymore.
- 6) P20: "...as glial activators (13.8%), glial inhibitors (2.6%)...". It is unclear what is meant by 100% here. Please specify.
- 7) P26: "... whereas inhibition of PDK2 may promote neuroprotective astrocytic migration toward injury sites." The authors should cite evidence that increased astrocyte migration is indeed neuroprotective. Faster migration could also be deleterious.
- 8) P29; the chemical formula of nitrite is NO₂⁻ (not NO₂). Please correct.
- 9) Cytotoxicity assay using LDH release: please specify how data was normalized. For instance, did the authors use fully lysed cells (e.g. by detergent) as 100% control?
- 10) P35: GAPDH was used as an internal control for real-time PCR. GAPDH is an enzyme of glycolysis and is not an appropriate

control given the fact that the authors show that glycolysis is a major target of regulation in their screen.

Reviewer #1 (Comments to the Authors (Required)):

In this manuscript titled "A Multiplexed siRNA screen identified key kinase signaling networks of brain glia" the authors are interrogating intracellular signaling pathways involved in glial cell activation, an understudied area. In order to do this, the researchers developed a multiplexed kinome wide siRNA screen to identify kinases that regulate inflammatory LPS induced phenotypes of glial populations. Authors used primary mixed glial cultures from embryonic mice, and four different phenotypic assays (Nitric oxide, cell viability/cytotoxicity, migration/wound healing, and phagocytosis/zymosan uptake). Authors also investigated role of ITK and PDK2 in microglial NO production/activation and astrocyte migration respectively in vivo in LPS and stab injury mouse models.

The work has important implications for our understanding of kinases regulating glial phenotypes in reactive states. However, several issues need to be addressed prior to publication:

Major issues

Comment 1: The work flow of multiplex siRNA screens should be described in more details. For instance based on Fig 1 authors tested 623 kinases using triplicate siRNA for each, with LPS or PBS treatment, and with all measurements performed in triplicates -to do this authors needed min of 117 96- well plates for one repetition. And to perform four different assays on the 117 wells authors needed total of 350 plates (two plates for supernatant for NO and viability, and original plates for migration and phagocytosis). This is a huge number of samples to process simultaneously and it will be worthy explaining how was this practically done as this could influence results. For example were all 623 kinases tested at the same time for simultaneous processing of 116 96 well plates or was this done over several independent experiments? How many litters of pups were used for all of the experiments? How many plates total were used? What quality checks were being run to ensure that conditions were the same across different plates/culture preparations?

Response: The kinome-wide siRNA screen was performed in a 96-well plate format. We used an siRNA library containing a pool of three siRNAs targeting each of the 623 kinases across seven plates. The seven library plates (master plates) were reformatted into 24 plates (daughter plates) to conduct the assay and obtain data in triplicates (by placing a single pool of kinase siRNAs into three wells) (Revised Fig 1). Mixed glial cells (MGCs) were seeded into 24 plates (assay plates) at 2,500 cells per well and transfected with the siRNA library, nontargeting siRNA, or transfection reagent alone. MGCs were then incubated with or without LPS (1 $\mu\text{g}/\text{mL}$) for 48 h (total 48 plates with 24 LPS-treated plates and 24 LPS-untreated plates). Following LPS stimulation, four different assays were conducted using either the culture medium or the remaining cells in the assay plates. All measurements were performed in triplicates. The whole multiplexed siRNA screen was repeated twice (96 plates in total). MGCs were prepared from the brains of 20 pups for a single round of screening. The LPS-untreated condition was used as the screening control (page 6). Quality checks across different plates or culture conditions were performed by including multiple controls, such as LPS-untreated mixed glial cells, nontargeting siRNA-transfected cells, and transfection reagent alone, in every assay plate. To determine the cellular proportion in different MGC preparations, immunocytochemical staining was performed using glia-specific markers (Fig S1; see our response to Comment 4).

[Figure removed by LSA Editorial Staff per authors' request]

Revised Figure 1. Overview of the multiplexed kinase siRNA screening strategy. For the kinome-wide siRNA screening, we used the siRNA mouse kinase library in a 96-well plate format. Each well contained a single pool of three different siRNA sequences targeting each of the 623 kinases (seven plates). The seven library plates (master plates) were reformatted to 24 plates (daughter plates) to conduct the assay and obtain data in triplicate. Mixed glial cells (MGCs) were seeded into 24 plates (assay plates) at a density of 2,500 cells per well and transfected with the siRNA library, nontargeting siRNA, or the transfection reagent alone. MGCs were then incubated with or without LPS (1 $\mu\text{g}/\text{mL}$) for 48 h (to a total of 48 plates with 24 LPS-treated plates and 24 LPS-untreated plates). Following LPS stimulation, four different assays were performed using either the culture medium or the remaining cells in the assay plates. All measurements were performed in triplicate. The whole multiplexed siRNA screen was repeated twice (96 plates in total). For a single screening round, MGCs were prepared from the brains of 20 pups. The LPS-untreated condition was used as screening control. See the “Materials and Methods” section for detailed descriptions of individual assays.

[Figure removed by LSA Editorial Staff per authors' request]

Revised Figure S1. Identification of astrocytes, microglia, and oligodendrocytes in mixed glial culture. MGC were isolated from whole brains of mouse pups at postnatal day 3. At 14 days *in vitro*, MGC were transferred to 96-well plates (2,500 cells per well). After 48 h, MGC were immunostained with anti-GFAP (1:1000; green), anti-Iba-1 (1:1000; gray), and anti-Olig2 (1:500; red) antibodies, for astrocytes, microglia, and oligodendrocytes, respectively. DAPI (blue) was used for counterstaining. Scale bar = 200 μm .

Page 6: The kinome-wide siRNA screen was performed in a 96-well plate format. We used an siRNA library containing a pool of three siRNAs targeting each of the 623 kinases across seven plates. The seven library plates were reformatted into 24 plates for the assay to obtain data in triplicate (by splitting a single pool of kinase siRNAs into three wells). MGCs, obtained from the brains of 20 pups for a single screening round, were seeded into the assay plates at a density of 2,500 cells per well and transfected with the siRNA library, nontargeting siRNA, or transfection reagent alone. MGCs were then incubated with or without LPS (1 $\mu\text{g}/\text{mL}$) for 48 h (to a total of 48 plates; 24 LPS-treated and 24 LPS-untreated plates). Following LPS stimulation, four different assays were performed using either the culture medium or the remaining cells in the assay plates. All measurements were performed in triplicate. The whole multiplexed siRNA screen was repeated twice (96 plates in total). The LPS-untreated condition was used as screening control. Quality checks across different plates or culture conditions were performed by including multiple controls, such as LPS-untreated MGCs, nontargeting siRNA-transfected cells, and transfection reagent alone, in every assay plate. To determine the cellular proportions in different MGC preparations, we performed immunocytochemical staining with glia-specific markers (Fig S1).

Comment 2: Authors should consider switching what is being used for main figures and supplementary figures. While the pathway analysis is interesting, the specific kinases and confirmation of those results would make the paper more easily understandable and more impactful. Therefore, showing the actual results of assays from the siRNA screen for the top kinases is highly recommended for Figs 2-5.

Response: Thank you for this comment. According to the reviewer's suggestion, we have rearranged supplementary figure 11 as the main figure 9. To reveal the top kinases for each assay, we revised figures 2–5.

[Figure removed by LSA Editorial Staff per authors' request]

Revised Figure 9. Schematic summary showing the relevance of microglial Itk and astrocytic Pdk2 in neuroinflammation and neurotoxicity. Brain injury results in the release or leakage of inflammatory molecules from the damaged brain tissue, activating the microglia (through Itk, asterisk). Aberrant microglial activation and phagocytosis can also induce neurotoxicity. This process is followed by astrocytic migration toward the injury site (through Pdk2 inhibition, asterisk). Reactive astrocytes can further cause glial scar formation. The astrocytic scar separates healthy tissue from the damaged tissue, exhibiting neuroprotective effects in brain injury. Thus, Itk inhibition may suppress neurotoxic microglial activation whereas Pdk2 inhibition may promote astrocytic migration and might assist with neurorepair.

[Figure removed by LSA Editorial Staff per authors' request]

Response Figure 1. Dual-flashlight plot for strictly standardized mean difference (SSMD) vs. fold-change (log₂ scale) in revised Fig 2–5. (A) Nitric oxide assay. (B) Cytotoxicity assay. (C) Wound healing assay. (D) Phagocytosis assay. Increased or decreased phenotypes were defined by fold-changes ≥ 0.5 or ≤ -0.5 , respectively (log₂ scale). Values on the x-axis indicate the average fold change of the three siRNAs for each target kinase. Final hit kinases were selected by SSMD values ≥ 1.65 or ≤ -1.65 . Several top kinases are indicated.

Comment 3: Authors use less strict statistical analysis for cytotoxicity presented in Fig 3 than the other 3 experiments. Authors need to use consistent method of analysis for all four phenotypes.

Response: Thank you for this comment. In response, we have reanalyzed our screen data using the same method (SSMD method combined with FC values) as in other screens. We have revised the main text (page 11), Fig 3, and Tables S6 and S7.

[Figure removed by LSA Editorial Staff per authors' request]

Revised Figure 3. Kinases that regulate glial cell death. (A) Dual-flashlight plot for strictly standardized mean difference (SSMD) vs. fold-change (log₂ scale) in the cytotoxicity assay. Increased or decreased phenotypes were defined by fold-changes ≥ 0.5 or ≤ -0.5 , respectively (log₂ scale). Values on the x-axis indicate the average fold-change of the three siRNAs for each target kinase. The selected hit kinases had SSMD values ≥ 1.65 or ≤ -1.65 . Top kinases are indicated. (B) Representative functional group network view for Gene Ontology (GO) terms generated using ClueGO. Terms were functionally grouped based on shared genes (kappa score) and are shown in different colors.

Page 11: Hits were selected using a dual-flashlight plot in which both the average FC and SSMD were considered simultaneously (Fig 3A). To select siRNA hits that increase cell death (kinases essential for survival), we used the following criteria: an average FC ≥ 0.5 (on the log₂ scale) and SSMD ≥ 1.65 . To select siRNA hits decreasing cell death (kinases enhancing glial cell death), we used an average FC ≥ -0.5 (on the log₂ scale) and SSMD ≥ -1.65 . Then, pharmacological validation was performed using commercially available small-molecule inhibitors for three kinases (Fig S5). Among them, Src kinase inhibitor (PP2) at 10 μ M enhanced cell death (Fig S5). Further, ClueGO analysis indicated that “neurotrophin TRK receptor signaling pathway” and “peptidyl-tyrosine autophosphorylation” were significantly enriched in the 15 kinases (Fig 3B). The most significant KEGG pathways enriched for glial survival kinases were related to the “thyroid hormone signaling pathway,” “inflammatory mediator regulation of TRP channels,” “MAPK signaling pathway,” and “Hh signaling pathway” (Table S7).

Minor:

Comment 4: IHC experiments with cultured microglia - it looks like there are very few microglia in images taken of cultured glial cells. Numbers wise, there are only ~20% microglia and ~70% astrocyte populations in these cultures. Is this consistent across different MGC preparations and different untreated/treated wells?

Response: We thank the reviewer for their comment. Yes, the cell-specific population of our culture was similar to that of a previous report (Giorgi-Coll et al., 2017) and statistically consistent across different preparations and untreated/treated conditions. We have added a table including detailed statistical information on the proportion of each glial cell type and interassay variation across different cultures or stimulation conditions (the interassay variation was < 20%, indicating consistency) (Li et al., 2017). The data have been stated in the revised main text (page 8) and added in Fig. S1 and Table S3.

[Figure removed by LSA Editorial Staff per authors' request]

Revised Figure S1. Identification of astrocytes, microglia, and oligodendrocytes in mixed glial culture. MGC were isolated from whole brains of mouse pups at postnatal day 3. At 14 days *in vitro*, MGC were transferred to 96-well plates (2,500 cells per well). After 48 h, MGC were immunostained with anti-GFAP (1:1000; green), anti-Iba-1 (1:1000; gray), and anti-Olig2 (1:500; red) antibodies, for astrocytes, microglia, and oligodendrocytes, respectively. DAPI (blue) was used for counterstaining. Scale bar = 200 μ m.

Table S3-1. Percentage of glial cell types in MGCs cultures.

Cell types	Cell markers	Preparation 1 ^a		Preparation 2		Preparation 3	
		PBS	LPS	PBS	LPS	PBS	LPS
Astrocyte	GFAP	63.4 \pm 2.0 ^b	58.5 \pm 4.1	67.4 \pm 1.5	62.4 \pm 4.0	66.6 \pm 2.2	72.1 \pm 0.8
Microglia	Iba-1	17.1 \pm 3.4	24.3 \pm 0.9	17.0 \pm 0.3	21.6 \pm 1.3	16.9 \pm 2.3	23.7 \pm 1.2
Oligodendrocyte	Olig2	14.9 \pm 0.3	12.5 \pm 0.6	16.0 \pm 2.5	12.0 \pm 0.5	13.1 \pm 0.4	14.2 \pm 0.4

^a MGCs prepared on different days

^b Percentage of each cell type was calculated. Total number of DAPI-positive cells was set to 100%.

Table S3-2. Interassay coefficient of variation among different MGCs preparations or between LPS-untreated and -treated wells.

Cell types	Preparations		LPS-untreated vs –treated wells		
	PBS	LPS	Preparation 1	Preparation 2	Preparation 3
Astrocyte	2.6 ^a	8.9	4.0 ^b	3.8	3.9
Microglia	0.5	4.9	17.3	11.6	16.8
Oligodendrocyte	8.1	7.4	8.8	10.8	4.0

^a Interassay coefficient of variation in percentage of glia cell type among different MGCs preparations

^b Interassay coefficient of variation in percentage of glia cell type between LPS-untreated and -treated wells

Page 8: Following immunocytochemical staining, the analysis of fluorescence images revealed the cellular composition of cultured MGCs (Fig S1; Table S3). MGCs comprised approximately 66% astrocytes, 15% microglia, and 15% oligodendrocytes in the LPS-untreated group and 64% astrocytes, 23% microglia, and 13% oligodendrocytes in the LPS-treated group. The interassay variation was <10% across different preparations or <20% between LPS-untreated and LPS-treated wells, indicating statistical consistency (Li et al., 2017).

Comment 5: Sup. Fig 2 - Western blot confirms the down regulation of specific kinases with the siRNAs in PBS condition. Authors should include LPS treatment and an LPS plus siRNA condition to determine whether LPS affects level of mRNA/protein reduction?

Response: We thank the reviewer for their comment. In response, we have conducted protein level validation and revised Supplementary Fig S3. Our data show that downregulation of specific kinases (mRNA/protein) with the siRNAs was similar in PBS and LPS conditions (All reduction rates in percentage were $P > 0.1$ in comparisons of PBS and LPS conditions). The data have been stated in the revised main text (page 8).

[Figure removed by LSA Editorial Staff per authors' request]

Revised Figure S3. Determination of siRNA knockdown efficiency in MGCs. (A)

Knockdown efficiency of siRNAs targeting 12 kinases at mRNA level determined using real-time PCR. Data are presented as mean \pm SEM (n = 3 replicate wells per group). Unpaired *t* test. **P* < 0.05 vs. control (Ctrl, without siRNAs) in the PBS-treated group. #*P* < 0.05 vs. Ctrl without siRNAs in the LPS-treated group. The kinases were selected based on phenotypic screening of siRNA hits (Table S2). **(B)** Confirmation of the knockdown at protein level. The kinases were selected based on inhibitory efficiency at mRNA level (< 50%, three kinases; > 50%, three kinases). Data are presented as mean \pm SEM (n = 3 replicate wells per group). Unpaired *t* test. **P* < 0.05 vs. Ctrl without siRNA. #*P* < 0.05 vs. Ctrl without siRNAs in the LPS-treated group. Assays (A and B) were repeated at least twice with similar results and the interassay coefficients of variation were < 20%, indicating a high level of confidence in the result.

Page 8: The efficiency of the siRNA-mediated knockdown of kinase gene expression was validated using real-time PCR and immunoblotting for the selected kinases. For validation at the mRNA level, 12 kinases were selected based on phenotypic screening of the siRNA library (Table S2); four from the activation screen; four from the migration screen; three from the phagocytosis screen; and four that did not show significant effects in any screens (Fig

S3A). For validation at the protein level, six kinases were selected based on inhibition efficiency at the mRNA level (>50%, three kinases; <50%, three kinases; Fig S3B). Our data showed similar siRNA downregulation of specific kinases (mRNA/protein) in PBS and LPS conditions: all reduction rates (%) had a *P* value of >0.1 in comparisons between PBS and LPS conditions.

Comment 6: Sup. Fig. 3-6 - Each of these supplemental figures has a table with information on whether a specific kinase is either up or down regulated. It would be helpful to make these table more quantitative with FC and or P value. That would make the information more impactful and valuable. Moreover, we strongly suggest data to be presented consistently as FC (i.e. Figure S3. please show as FC instead of nitrite concentration) to be comparable with siRNA data.

Response: We thank the reviewer for their comment. In response to this suggestion, we have revised Figs S4–S8 by adding the FC values.

[Figure removed by LSA Editorial Staff per authors' request]

Revised Figure S4. Pharmacological validation of the kinase hits from the siRNA screen (NO assay). Mixed glial cell cultures were treated with LPS (1 µg/mL) and commercially available small-molecule kinase inhibitors as indicated. After 48 h, the NO and MTT assays were performed. The results were compared with those obtained from the siRNA screen dataset. The NO assay results are presented as fold change (\log_2 FC); ns indicates no statistical difference relative to each control (without siRNA or vehicle). Blue indicates a decrease compared with each control. The effect sizes (Hedges' *g* values) are shown as well.

[Figure removed by LSA Editorial Staff per authors' request]

Revised Figure S5. Validation of the kinase hits identified in the cytotoxicity assay. Pharmacological validation of the kinase hits from the siRNA screen (cytotoxicity assay).

Mixed glial cell cultures were treated with LPS (1 $\mu\text{g}/\text{mL}$) and commercially available small-molecule kinase inhibitors as indicated. After 48 h, the MTT assay was conducted. The results were compared with those obtained from the siRNA screen dataset. Cytotoxicity is represented as fold change ($\log_2 \text{FC}$) or not significantly (ns) different relative to each control (without siRNA or vehicle). The red color indicates an increase compared with each control. The effect sizes (Hedges' g values) are shown as well.

[Figure removed by LSA Editorial Staff per authors' request]

Revised Figure S6. Validation of the kinase hits identified in the wound healing assay.

Pharmacological validation of the kinase hits from siRNA screening (migration assay). Mixed glial cells were treated with LPS (1 $\mu\text{g}/\text{mL}$) and commercially available small-molecule kinase inhibitors as indicated. After 48 h, a wound-healing assay was conducted. The results were compared with those obtained from the siRNA screen dataset. Glial migration is presented as fold change ($\log_2 \text{FC}$); ns indicates no statistical difference relative to each control (without siRNA or vehicle). Red and blue indicate an increase or decrease, respectively, compared with each control. The effect sizes (Hedges' g values) are shown as well.

[Figure removed by LSA Editorial Staff per authors' request]

Response Figure 2 (Revised Figure S7). Comparison of glial phenotypes after stimulation with DAMPs (A, B) and the different neuroinflammatory aspects altered by kinase knockdown (C). Comparison of glial nitric oxide (NO) production (A) and migration (B) after treatment with LPS or damage-associated molecular patterns (DAMPs). Mixed glial cell (MGC) cultures were transfected overnight with 12 kinase siRNAs. After 48 h, LPS (1 $\mu\text{g}/\text{mL}$), HMGB1 protein (20 $\mu\text{g}/\text{mL}$), or MGC lysate (1 $\mu\text{g}/\text{mL}$) were applied. (C)

Comparison of different neuroinflammatory aspects altered by kinase knockdown. Values are represented as fold-change (\log_2 FC) or not significantly different (ns) relative to the control (without siRNA). Red and blue indicate an increase or decrease, respectively, compared with each control.

[Figure removed by LSA Editorial Staff per authors' request]

Response Figure 3 (Revised Figure S8). Comparison of glial phenotypes between MGC and single cell type cultures. Comparison of LPS-induced NO production (A) and migration (B) among mixed glial cells, microglia, and astrocyte cultures after transfection with kinase siRNAs. At 14 days *in vitro*, astrocytes were isolated from MGC by shaking overnight to remove other cell types. Microglia were isolated by mild trypsin treatment. Cells were transfected with 12 kinase siRNAs overnight. After 48 h, LPS (1 μ g/ml) was added. Values are represented as fold-change (\log_2 FC) or not significantly different (ns) relative to the control (without siRNA). Red and blue indicate an increase or decrease, respectively, compared with each control.

Comment 7: In Sup Fig 3 authors should use two way ANOVA to determine effect of LPS and kinase inhibitors.

Response: We thank the reviewer for pointing out our typo. We have corrected the typo in the revised Fig S4.

[Figure removed by LSA Editorial Staff per authors' request]

Revised Figure S4. Validation of the kinase hits identified in the nitric oxide assay. (A)

The Toll-like receptor 4 signaling pathway and kinases identified as nitric oxide (NO) production regulators in this study. Asterisks and numbers indicate log₂-fold changes in the NO assay. **(B)** Pharmacological validation of the kinase hits from the siRNA screen (NO assay). Mixed glial cell cultures were treated with LPS (1 µg/mL) and commercially available small-molecule kinase inhibitors as indicated. After 48 h, the NO and MTT assays were performed. The results were compared with those obtained from the siRNA screen dataset. The NO assay results are presented as fold-change (log₂ FC); ns indicates no statistical difference relative to each control (without siRNA or vehicle). Blue indicates a decrease compared with each control. The effect sizes (Hedges' *g* values) are shown as well. Data are presented as mean ± SEM (n = 3 replicate wells per group). Two-way ANOVA was followed by Tukey's post-*hoc* test. **P* < 0.05. Veh, vehicle. Gray, PBS-treated group; black, LPS-treated group. The assay was repeated at least twice with similar results and the interassay coefficients of variation were <20%, indicating a high level of confidence in the results.

Comment 8: When calculating Hedges effect size what n was used? For instance each kinase had triplicate siRNA -were all three siRNA was used? Did all three show same results? Please show all data points in histograms (a dot for each biological or technical replicate).

Response: To calculate the Hedges effect size in the siRNA effect, we obtained data from three replicate siRNA wells (n = 3 technical replicates) and two different preparations of mixed glial cells (n = 2 independent experiments). Each well contained a pool of three siRNAs mixture targeting each of the kinases. The results represent the combined effects of the 3 siRNAs targeting single kinase. For inhibitor experiments, data were similarly obtained from three technical replicates and two different preparations of mixed glial cells (n = 2 independent experiments). We have added all the data points in the revised Fig S4 and S6.

[Figure removed by LSA Editorial Staff per authors' request]

Revised Figure S4. Validation of the kinase hits identified in the nitric oxide assay.

Pharmacological validation of the kinase hits from the siRNA screen (NO assay). Mixed glial cell cultures were treated with LPS (1 $\mu\text{g}/\text{mL}$) and commercially available small-molecule kinase inhibitors as indicated. After 48 h, the NO and MTT assays were performed. The results were compared with those obtained from the siRNA screen dataset. The NO assay results are presented as fold-change (\log_2 FC); ns indicates no statistical difference relative to each control (without siRNA or vehicle). Blue indicates a decrease compared with each control. The effect sizes (Hedges' g values) are shown as well. Data are presented as mean \pm SEM (n = 3 replicate wells per group). Two-way ANOVA was followed by Tukey's post-*hoc* test. * $P < 0.05$. Veh, vehicle. Gray, PBS-treated group; black, LPS-treated group. The assay was repeated at least twice with similar results and the interassay coefficients of variation were $<20\%$, indicating a high level of confidence in the results.

[Figure removed by LSA Editorial Staff per authors' request]

Revised Figure S6. Validation of the kinase hits identified in the wound healing assay.

Pharmacological validation of the kinase hits from siRNA screening (migration assay). Mixed glial cells were treated with LPS (1 µg/mL) and commercially available small-molecule kinase inhibitors as indicated. After 48 h, a wound-healing assay was conducted. The results were compared with those obtained from the siRNA screen dataset. Glial migration is presented as fold-change (\log_2 FC); ns indicates no statistical difference relative to each control (without siRNA or vehicle). Red and blue indicate an increase or decrease, respectively, compared with each control. The effect sizes (Hedges' g values) are shown as well. Data are presented as mean \pm SEM ($n = 3$ replicate wells per group). One-way ANOVA was followed by Tukey's post-hoc test. * $P < 0.05$ vs. Veh (vehicle). The assay was repeated at least twice with similar results and the interassay coefficients of variation were $<20\%$, indicating a high level of confidence in the results.

Comment 9: Authors state in results on NO production "Overall, we identified 16 kinases that inhibited glial activation (glial inhibitors) and 86 kinases that enhanced glial activation (glial activators)." Indicating that most identified kinases were glial activators/NO-increasing. Later in the same paragraph authors state "Due to the limited number of NO increasing kinases identified in the screen (15.7%), the selected hits were coincidentally either NO-decreasing kinases or kinases without significant effect", suggesting that most were glial inhibitors/NO-decreasing. Could authors please clarify these contradicting sentences?

Response: We thank the reviewer for pointing out our mistake. We have clarified these sentences in the revised manuscript (page 10).

Page 10: Due to the limited number of NO-decreasing kinases identified in the screen (15.7%), the selected hits were coincidentally either NO-increasing kinases or kinases without significant effects.

Comment 10: For survival data presented in Sup Fig 3 could authors create table similar to NO (comparing effects of siRNA screen and inhibitors)?

Response: We thank the reviewer for their comment. In response, we have added the table in the revised Fig S5.

[Figure removed by LSA Editorial Staff per authors' request]

Revised Figure S5. Validation of the kinase hits identified in the cytotoxicity assay.

Pharmacological validation of the kinase hits from the siRNA screen (cytotoxicity assay). Mixed glial cell cultures were treated with LPS (1 $\mu\text{g}/\text{mL}$) and commercially available small-molecule kinase inhibitors as indicated. After 48 h, the MTT assay was conducted. The results were compared with those obtained from the siRNA screen dataset. Cytotoxicity is represented as fold-change (\log_2 FC) or not significantly (ns) different relative to each control (without siRNA or vehicle). The red color indicates an increase compared with each control. The effect sizes (Hedges' g values) are shown as well. Data are presented as mean \pm SEM ($n = 3$ replicate wells per group). One-way ANOVA followed by Tukey's post-*hoc* test. $*P < 0.05$ vs. vehicle (Veh). The assay was repeated at least twice with similar results and the interassay coefficients of variation were $< 20\%$, indicating a high level of confidence in the results.

Comment 11: Could authors please clarify why in Fig 6E there is 2 fold increase in PLC phosphorylation 1hr vs 30 min in PBS? What causes this increase in PLC phosphorylation in PBS?

Response: We thank the reviewer for their comment. The increase in PLC phosphorylation in the PBS group may be due to an artifact in the BV-2 microglial cell line in the long-term culture. The results were replaced by new data obtained from fresh primary microglia culture.

[Figure removed by LSA Editorial Staff per authors' request]

Revised Figure 6E. Immunoblots of kinases related to the T cell receptor signaling pathway. Primary microglia were treated with LPS (1 $\mu\text{g}/\text{mL}$). Total protein was harvested at the

indicated time points and subjected to immunoblot analysis (*upper*) and quantification (*lower*). p-PLC- γ , phosphorylated PLC- γ ; PLC- γ , total PLC- γ ; p-Vav, phosphorylated Vav1; Vav1, total Vav1. Data are presented as mean \pm SEM (n = 3 replicate wells per group). Unpaired *t*-test. **P* < 0.05

Comment 12: Fig 7 should show the effects of ITK inhibition on PLC Phosphorylation in vivo.

Response: We thank the reviewer for their comment. We have added these data to the revised Figure 7. This was stated in the revised manuscript (page 18).

[Figure removed by LSA Editorial Staff per authors' request]

Revised Figure 7E. PLC- γ activity *in vivo*. PLC- γ immunoblots in hippocampal tissues at 24 h after LPS injection (5 mg/kg, i.p.) with or without the Itk inhibitor BMS509744 (10 mg/kg, i.p.). The quantification is shown in the adjacent graph (n = 3 replicates per group). Data are presented as mean \pm SEM. One-way ANOVA followed by Tukey's post-*hoc* test. **P* < 0.05.

Page 18: LPS-induced PLC- γ phosphorylation in the brain was significantly blocked by Itk inhibitor (Fig 7E).

Comment 13: Could authors please explain the differences in the extent of LPS induced PLC phosphorylation in BV-2 cells (i.e. in Fig 6E it is 2 fold after 1 hr., and in Fig 7B it is 20 fold after 1 hr). This is especially confusing as figure legends states". The assay was repeated at least twice with similar results and the inter-assay coefficients of variation was < 20%, indicating a high level of confidence in the result". Yet these results indicate 10 fold difference in inter-assays.

Response: We thank the reviewer for their comment. The difference in LPS-induced PLC phosphorylation in Fig 6E and 7B was due to a difference in the contrast of gel images when we measured the intensity of the bands. The results were replaced by PLC phosphorylation data obtained from primary microglia cultures.

[Figure removed by LSA Editorial Staff per authors' request]

Revised Figure 6E. Immunoblots of kinases related to the T cell receptor signaling pathway. Primary microglia were treated with LPS (1 $\mu\text{g}/\text{mL}$). Total protein was harvested at the indicated time points and subjected to immunoblot analysis (*upper*) and quantification (*lower*). p-PLC- γ , phosphorylated PLC- γ ; PLC- γ , total PLC- γ ; p-Vav, phosphorylated Vav1; Vav1, total Vav1. Data are presented as mean \pm SEM (n = 3 replicate wells per group). Unpaired *t* test. **P* < 0.05. The assay was repeated at least twice with similar results and the interassay coefficients of variation was <20%, indicating a high level of confidence in the result.

[Figure removed by LSA Editorial Staff per authors' request]

Revised Figure 7B. Immunoblots of PLC- γ (an Itk substrate) after treatment of primary microglia with LPS (1 $\mu\text{g}/\text{mL}$) in the presence or absence of the Itk inhibitor BMS509744 (1 $\mu\text{g}/\text{mL}$). The quantification is shown in the adjacent graph (n = 3 replicate wells per group). Data are presented as mean \pm SEM. One-way ANOVA followed by Tukey's post-*hoc* test. **P* < 0.05.

Comment 14: Could authors discuss why in the stab injury microglia do not migrate to the injury site (Figure S8C)?

Response: We thank the reviewer for pointing out our mistake. We have now added the results of statistical analysis to the revised Fig S10C. Our data revealed statistically significant differences in the microglial cell numbers in different areas under control condition (gray bars in the graph below), indicating actual microglial migration toward the injury site.

[Figure removed by LSA Editorial Staff per authors' request]

Revised Figure S10C. Pdk2 inhibition does not significantly affect microglial migration *in vivo*. Male C57BL/6 mice were intracortically injected with vehicle (control, Ctrl) or AZD7545 (1 μ M). *Pdk2* KO mice were injured through needle stabbing. BrdU was administered intraperitoneally every 6 h, as indicated. At the end of the experiment, the mice were euthanized, and brain sections were obtained in the transverse plane. The quantification of Iba-1-positive microglia around the injection site in the prefrontal cortex is shown in an adjacent graph. The cell numbers in each concentric circle (radius step size = 500 μ m) were analyzed. Data are presented as mean \pm SEM (n = 3 mice per group). One-way ANOVA was followed by Tukey's post-*hoc* test. * P < 0.05 vs. control (Ctrl) in area 1. Abbreviations: ns, not significant.

Comment 15: Sup. Fig. 9 and 10 - These figures would be far more impactful if we knew what kinases fell into each cluster and each category. Having a table of that information along with the Venn diagrams and pie charts would make the importance of these figures much more clear.

Response: We thank the reviewer for their comment. Suggested information along with Venn diagrams and pie charts have been provided in Tables S12 and S13.

Comment 16: Sup. Fig 11 - this figure should be in the main body of the paper. This is the main conclusion the paper is trying to make, it shouldn't be relegated to the supplemental.

Response: We thank the reviewer for their comment. In response, we have moved Fig S11 to the main body of the paper as figure 9.

Reviewer #2 (Comments to the Authors (Required)):

The MS by Kim et al presents a novel screen for kinases which are involved in neuroinflammatory processes in glial cells. They use mixed glial cell cultures, which in their hands mainly contain astrocytes and microglia and knock down a total of 623 kinases using siRNA. These cultures are incubated with LPS to mimic an inflammatory event. Four different parameters are assessed in the following which correlate with different aspects of neuroinflammation: 1) NO production via detection of nitrite; 2) cell death via release of LDH; 3) glial migration using a wound healing assay; 4) phagocytosis using labelled zymosan as substrate. The authors identify numerous kinases, which are involved in the different processes. The relevance of few selected kinases is studied in cell culture as well as in vivo experiments using pharmacology and genetic approaches. Two kinases are studied in more detail: Interleukin-2-inducible T-cell kinase (ITK) is shown to be involved in neuroinflammation as its KD affects the NO production, which is substantiated by additional experiments. PDK2 is shown to modulate astrocyte metabolism, and astrocyte migration. The authors show that PDK2 KO leads to increased oxygen consumption which is interpreted as a switch from glycolysis to Oxphos. This notion could be substantiated rather easily by measuring glycolytic rate using the same system (Seahorse), which then should show a reduced glycolytic rate. However, a causal connection of the change metabolism and reduced migration (as suggested by the authors) is not shown by experimental data. It might be independent processes regulated by the same kinase(s). This should be more carefully stated and discussed, or -alternatively- the causal relation should be shown experimentally.

Additional issues which should be addressed:

Comment 1: The author state that their cultures contain astrocytes (71.5%) and microglia (27.5%). They should at least comment that e.g. oligodendrocytes are lacking, which have been shown to be involved in neuroinflammation as well. Furthermore, no information is given on how many cultures were quantified and how large the variability of these cultures is. Therefore, more detailed statistical information needs to be given here. This also applies to the statement on transfection efficiency: "51.1% and 72.7% transfection rates".

Response: We thank the reviewer for their comment. In response, we measured the proportions of astrocytes, microglia, and oligodendrocytes in our MGC culture system. While mixed glial cultures consisted of approximately 66% astrocytes, 15% microglia, and 15% oligodendrocytes in the untreated group, cellular proportions of 64% astrocytes, 23% microglia, and 13% oligodendrocytes were observed in the LPS-treated group. Data were obtained from three cell culture wells (n = 3 technical replicates) and three different preparations of mixed glial cells (n = 3 independent experiments). The interassay variation was < 20%, indicating statistical consistency between the cultures. In this study, we focused on the phenotypes of astrocytes and microglia—the main players in neuroinflammation. Transfection rates of astrocytes (70%) and microglia (54.8%) in the untreated group, and transfection rates of astrocytes (67.1%) and microglia (56.3%) in the LPS-treated group were observed. The interassay variation in transfection rate was < 20% between the untreated and LPS-treated groups. We have stated this in the revised main text (page 8) and added a table with detailed statistical information on the proportion of glial cells, interassay variation across different cultures, and transfection efficiency (Table S3).

Table S3-1. Percentage of glial cell types in MGCs cultures.

Cell types	Cell markers	Preparation 1 ^a		Preparation 2		Preparation 3	
		PBS	LPS	PBS	LPS	PBS	LPS
Astrocyte	GFAP	63.4±2.0 ^b	58.5±4.1	67.4±1.5	62.4±4.0	66.6±2.2	72.1±0.8
Microglia	Iba-1	17.1±3.4	24.3±0.9	17.0±0.3	21.6±1.3	16.9±2.3	23.7±1.2
Oligodendrocyte	Olig2	14.9±0.3	12.5±0.6	16.0±2.5	12.0±0.5	13.1±0.4	14.2±0.4

^a MGCs prepared on different days

^b Percentage of each cell type was calculated. Total number of DAPI-positive cells was set to 100%.

Table S3-2. Interassay coefficient of variation among different MGCs preparations or between LPS-untreated and -treated wells.

Cell types	Preparations		LPS-untreated vs -treated wells		
	PBS	LPS	Preparation 1	Preparation 2	Preparation 3
Astrocyte	2.6 ^a	8.9	4.0 ^b	3.8	3.9
Microglia	0.5	4.9	17.3	11.6	16.8
Oligodendrocyte	8.1	7.4	8.8	10.8	4.0

^a Interassay coefficient of variation in percentage of glia cell type among different MGCs preparations

^b Interassay coefficient of variation in percentage of glia cell type between LPS-untreated and -treated wells

Table S3-3. Transfection efficiency in MGCs and interassay variation across different cultures.

Cell types	Cell markers	Preparation 1		Preparation 2		Preparation 3		Interassay CV	
		PBS	LPS	PBS	LPS	PBS	LPS	PBS	LPS
Astrocyte	GFAP	72.4 ± 9.5 ^a	74.3 ± 7.9	65.6 ± 7.6	71.0 ± 3.9	72.2 ± 2.5	70.1 ± 5.7	4.5 ^b	3.5
Microglia	Iba-1	50.5 ± 11.0	51.9 ± 7.2	50.8 ± 4.8	50.8 ± 8.4	54.0 ± 1.5	56.1 ± 8.4	7.0	0.8
Oligodendrocyte	Olig2	58.5 ± 20.2	59.6 ± 9.5	72.6 ± 7.6	49.9 ± 3.0	68.5 ± 3.6	51.0 ± 6.5	8.9	8.1

^a Percentage of transfection efficiency was calculated as follows: (the number of transfected cells/the number of total cells stained with each cell marker) x 100

^b Coefficient of variation (CV) in percentage among different cell preparations.

Page 8: Following immunocytochemical staining, the analysis of fluorescence images revealed the cellular composition of cultured MGCs (Fig S1; Table S3). MGCs comprised approximately 66% astrocytes, 15% microglia, and 15% oligodendrocytes in the LPS-untreated group and 64% astrocytes, 23% microglia, and 13% oligodendrocytes in the LPS-treated group. The interassay variation was <10% across different preparations or <20% between LPS-untreated and LPS-treated wells, indicating statistical consistency (Li et al., 2017). In this study, we focused on the phenotypes of astrocytes and microglia—the main players in neuroinflammation. The efficiency of siRNA transfection was determined using a Cy3-labeled siRNA transfection control [siRNA of glyceraldehyde 3-phosphate dehydrogenase (GAPDH)] followed by immunocytochemical staining with antibodies against

glial cell markers, such as anti-glial fibrillary acidic protein (GFAP) or anti-ionized calcium binding protein (Iba-1; Fig S3C). Transfection rates of astrocytes and microglia (70.1% and 54.8%, respectively) in the LPS-untreated group and of those (67.1% and 56.3%, respectively) in the LPS-treated group were determined, and the intergroup variation in transfection rate was found to be < 20% (Table S3).

Comment 2: In general, it is hard to understand why in some cases a subset of kinases is selected. E.g. in Fig S6, 12 kinases are studied in panels A, B, but only 5 in panel C. A rationale and the criteria for these selections (not only for Fig. S6 but throughout the paper) would be helpful.

Response: We selected a subset of kinases (12 kinases) based on the siRNA phenotypic screening results (Table S2). Three or four kinases were chosen from the results of each phenotypic screen: 4 kinases from activation screen; 4 kinases from migration screen; 3 kinases from phagocytosis screen; 4 kinases that did not show significant effects in any screens (page 8). This information has been added to the Table S2. In panel C of Fig S6 (revised Fig S8), the top 5 kinases identified in MGC cytotoxicity assay were selected for validation experiments with cultures of a single cell type (page 16). The selection criteria for the 6 kinases in panel B of Fig S2 (validation for protein expression) were as follows: the kinases were selected on the basis of siRNA knockdown efficiency at the mRNA level (>50%, three kinases; <50%, three kinases). Pharmacological validation for 24 kinases were conducted using commercially available kinase inhibitors (page 10).

Table S2. Criteria for selecting a subset of kinases used in the validation experiments of siRNA knockdown efficiency. Three or four kinases were chosen from the results of each phenotypic screen: 4 kinases from activation screen; 4 kinases from migration screen; 3 kinases from phagocytosis screen; 4 kinases that did not show significant effects in any screens. Some kinases showed significant effects in two different screens.

Targets	Results of glial phenotypic screen			
	No significant effects in any screens	Significant effects in activation screen	Significant effects in migration screen	Significant effects in phagocytosis screen
Epha1	○			
Epha2	○			
Akt2	○			
Mlkl	○			
Itk		○		
Ckb		○		
Irak4		○		
Ulk1		○		○
Pdk2			○	
Pkm2			○	
Lrrk2			○	○
Frap1			○	○

[Figure removed by LSA Editorial Staff per authors' request]

Revised Figure S3. Determination of siRNA knockdown efficiency in MGCs. (A) Knockdown efficiency of siRNAs targeting 12 kinases at mRNA level determined using real-time PCR. Data are presented as mean \pm SEM (n = 3 replicate wells per group). Unpaired *t* test. **P* < 0.05 vs. control (Ctrl, without siRNAs) in the PBS-treated group. #*P* < 0.05 vs. Ctrl without siRNAs in the LPS-treated group. The kinases were selected based on phenotypic screening of siRNA hits (Table S2). **(B)** Confirmation of the knockdown at protein level. The kinases were selected based on inhibitory efficiency at mRNA level (< 50%, three kinases; > 50%, three kinases). Data are presented as mean \pm SEM (n = 3 replicate wells per group). Unpaired *t* test. **P* < 0.05 vs. Ctrl without siRNA. #*P* < 0.05 vs. Ctrl without siRNAs in the LPS-treated group. Assays (A and B) were repeated at least twice with similar results and the interassay coefficients of variation were < 20%, indicating a high level of confidence in the result.

Page 8: The efficiency of the siRNA-mediated knockdown of kinase gene expression was validated using real-time PCR and immunoblotting for the selected kinases. For validation at the mRNA level, 12 kinases were selected based on phenotypic screening of the siRNA library (Table S2); four from the activation screen; four from the migration screen; three from the phagocytosis screen; and four that did not show significant effects in any screens (Fig S3A). For validation at the protein level, six kinases were selected based on inhibition efficiency at the mRNA level (>50%, three kinases; <50%, three kinases; Fig S3B).

Page 10: Next, the effects of siRNAs on NO production were pharmacologically confirmed by re-assaying with commercially available small-molecule inhibitors for 24 kinases (Fig S4B).

Page 16: Further, we assessed cell death in cultures of a single cell type (microglia vs. astrocytes). Then, the top five kinases identified in MGC cytotoxicity assay (~ 50% cell death) were selected for validation experiments with cultures of a single cell type (Fig S8C).

Comment 3: Experiments shown in Fig. 7C and corresponding text: The number of microglia increases by a factor of 3, the number of astrocytes by a factor of 2 when exposed to LPS. The authors state that this is due to "activation". However, such an increase in cell numbers most likely includes massive proliferation, as microglia are Iba1+ even in the quiescent state and -in the hippocampus, which is studied here- also most astrocytes are GFAP+ under basal conditions.

Response: We thank the reviewer for their comment. We agree that glial cell number may represent their proliferation under pathological conditions. To measure glial activation that is better characterized by hypertrophy or termed "gliosis" (Pekny and Nilsson, 2005; Silver and Miller, 2004), we evaluated changes in the intensity of GFAP and Iba-1 staining, which reflect an increase in glial cell size or proliferation. The results were replaced by the new data in the revised Figure 7C and stated in the revised manuscript (page 17).

[Figure removed by LSA Editorial Staff per authors' request]

Revised Figure 7C. Inhibition of microglial Itk alleviates LPS-induced neuroinflammation. Immunohistochemistry of glial cells in the hippocampal region. Mice were intraperitoneally (i.p.) injected with LPS (5 mg/kg) with or without BMS509744 (10 mg/kg, i.p.). Astrocytes and microglia in the hippocampus were immunostained with anti-GFAP and anti-Iba-1 antibodies, respectively, at 24 h after LPS treatment. Scale bar = 400 μ m. The quantification of glial activation is shown in the adjacent graph (n = 5 mice per group; each data point indicates the average value of six fields of view per mouse).

Page 17: Next, we sought to determine the role of Itk in microglia-mediated neuroinflammation. In the systemic LPS injection-induced neuroinflammation model, co-administration of the Itk inhibitor BMS509744 (10 mg/kg) significantly reduced the activation of microglia and astrocytes. Glial activation characterized by hypertrophy or

“gliosis” (Pekny and Nilsson, 2005; Silver and Miller, 2004) was measured based on the immunofluorescence intensity of Iba-1-positive microglia and GFAP-positive astrocytes (Fig. 7C).

Comment 4: Hexokinase, pyruvate kinase M, and pyruvate dehydrogenase kinase isoform 2 are identified as most potent kinases in the deceleration of glial migration. The authors state that these are "associated with glycolytic metabolism". This is essentially true, but it might be helpful to mention that hexokinase and pyruvate kinase M are actually enzymes within the glycolytic pathway which do not phosphorylate proteins as regulatory kinases.

Response: As the reviewer correctly indicated, hexokinase and pyruvate kinase M (PKM) are indeed major glycolytic enzymes and do not directly regulate the phosphorylation of other proteins. However, there are studies showing that these enzymes may indirectly regulate protein phosphorylation (Roberts and Miyamoto, 2015; Zhang et al., 2019). We have discussed this issue in the revised main text (page 19).

Page 19: All these kinases are closely associated with the glycolytic metabolism. Hexokinase and PKM may indirectly regulate the phosphorylation of other proteins (Roberts and Miyamoto, 2015; Zhang et al., 2019), although these enzymes do not directly phosphorylate proteins as regulatory protein kinases.

Comment 5: When using the glycolysis inhibitor 2DG, the authors need to indicate if there are any other substrates in addition to glucose within the medium, and -if so- which ones. If 2DG is used in the presence of glucose only, also mitochondria are depleted from their substrates as pyruvate is not produced from glycolysis anymore.

Response: We used the Seahorse XF basal DMEM (Response Figure 1), containing 1.5 mM sodium pyruvate, 1 mM glutamine, and 25 mM glucose, for the assay. We have added this information in the Materials and Methods section (page 37).

[Figure removed by LSA Editorial Staff per authors' request]

Response Figure 1. Composition of XF basal DMEM.

Page 37: Briefly, astrocytes from WT and *Pdk2* KO mice were plated at a density of 40,000 cells per well with LPS in a Seahorse XF24 plate and cultured for 16 h. One hour before the assay, the medium was exchanged for Seahorse XF basal DMEM containing 1.5 mM sodium pyruvate, 1 mM glutamine, and 25 mM glucose. Rotenone or 2-DG, FCCP, and oligomycin were diluted in XF24 medium with LPS and loaded into the accompanying cartridge to achieve final concentrations of 2, 1, and 1 $\mu\text{g/mL}$, respectively.

Comment 6: P20: "...as glial activators (13.8%), glial inhibitors (2.6%)...". It is unclear what is meant by 100% here. Please specify.

Response: We have revised the main text for clarification (Page 21).

Page 21: We used a systematic and stringent screening approach and identified the essential kinases, among the 623 kinases (100%), acting as glial activators (13.8%) or inhibitors (2.6%), key regulators of glial cell death/survival (2.4%), glial migration accelerators (12%) or inhibitors (3.5%), and phagocytosis enhancers (9.1%) or suppressors (19.9%). Validation experiments using pharmacological inhibitors of the representative kinases revealed phenotypic changes similar to those identified in the multiplex kinase siRNA screen.

Comment 7: P26: "... whereas inhibition of PDK2 may promote neuroprotective astrocytic migration toward injury sites." The authors should cite evidence that increased astrocyte migration is indeed neuroprotective. Faster migration could also be deleterious.

Response: We thank the reviewer for their comment. We have revised the main text with additional references (page 27).

Page 27: Our findings suggest that Itk and Pdk2 are potential kinase targets for therapeutic modulation of the neuroinflammatory phenotypes of glia (Fig 9). Itk inhibition may suppress neurotoxic microglial activation whereas Pdk2 inhibition may promote astrocytic migration, aiding neurorepair (Chiareli et al., 2021; Renault-Mihara et al., 2011). Combination therapies targeting these two kinases may exhibit enhanced therapeutic effects.

Comment 8: P29; the chemical formula of nitrite is NO_2^- (not NO_2). Please correct.

Response: We thank the reviewer for pointing out our mistake. We have corrected this error accordingly (page 30).

Page 30: NO assay. The NO_2^- concentration in culture media was measured to assess NO production in MGCs using the Griess reagent.

Comment 9: Cytotoxicity assay using LDH release: please specify how data was normalized. For instance, did the authors use fully lysed cells (e.g. by detergent) as 100% control?

Response: In our cytotoxicity assay, the data were calculated using the following formula:

percentage cytotoxicity = $100 \times (\text{experimental LDH release}/\text{maximum LDH release})$ according to the manufacturer's instructions. Lysis solution (0.8% Triton X-100) was used to generate the maximum LDH release with 100% cell death control. The final data are represented as cell survival in the original Supplementary Table 4. We have reanalyzed the cytotoxicity using fold-change and SSMD (Revised Fig 3) (page 30).

Page 30: Cytotoxicity assay. Cytotoxicity was evaluated by measuring the amount of released LDH using the CytoTox 96 Non-Radioactive Cytotoxicity Assay Kit (Promega, Madison, WI, USA), according to manufacturer's instructions. Culture media (50 μL) from the MGCs were incubated with an LDH substrate solution (50 μL) for 20 min in a dark room. After adding the stop solution, the absorbance was measured at 490 nm using a microplate reader (SpectraMax M5; Molecular Devices). The data were calculated using the following formula: % cytotoxicity = $100 \times (\text{experimental}/\text{maximum LDH release})$, according to manufacturer's instructions. Lysis solution (0.8% Triton X-100) was used to generate the maximum LDH release with 100% cell death control.

Comment 10: P35: GAPDH was used as an internal control for real-time PCR. GAPDH is an enzyme of glycolysis and is not an appropriate control given the fact that the authors show that glycolysis is a major target of regulation in their screen.

Response: We thank the reviewer for their comment. In response, we reanalyzed real-time PCR data using the two other internal controls, Rplp0 and Tub1a, as well as a model-based variance and stability calculation for normalization (Andersen et al., 2004; Vandesompele et al., 2002) (page 36).

Page 36: Real-time PCR was performed using a One-Step SYBR PrimeScript RT-PCR Kit (Perfect Real-Time; Takara Bio), followed by detection using an ABI Prism 7000 Sequence Detection System (Applied Biosystems). Normalization was performed using two internal controls, ribosomal protein lateral stalk subunit P0 (Rplp0) and tubulin alpha 1A (Tub1a), as well as a model-based variance and stability calculation (Andersen et al., 2004; Vandesompele et al., 2002).

References

- Andersen, C.L., J.L. Jensen, and T.F. Orntoft. 2004. Normalization of real-time quantitative reverse transcription-PCR data: a model-based variance estimation approach to identify genes suited for normalization, applied to bladder and colon cancer data sets. *Cancer Res.* 64:5245-5250.
- Chiareli, R.A., G.A. Carvalho, B.L. Marques, L.S. Mota, O.C. Oliveira-Lima, R.M. Gomes, A. Birbrair, R.S. Gomez, F. Simao, F. Klempin, M. Leist, and M.C.X. Pinto. 2021. The Role of Astrocytes in the Neurorepair Process. *Front Cell Dev Biol.* 9:665795.
- Giorgi-Coll, S., A.I. Amaral, P.J.A. Hutchinson, M.R. Kotter, and K.L.H. Carpenter. 2017. Succinate supplementation improves metabolic performance of mixed glial cell cultures with mitochondrial dysfunction. *Sci Rep.* 7:1003.
- Li, N., S. Katz, B. Dutta, Z.L. Benet, J. Sun, and I.D. Fraser. 2017. Genome-wide siRNA screen of genes regulating the LPS-induced NF-kappaB and TNF-alpha responses in mouse macrophages. *Sci Data.* 4:170008.
- Pekny, M., and M. Nilsson. 2005. Astrocyte activation and reactive gliosis. *Glia.* 50:427-434.
- Renault-Mihara, F., H. Katoh, T. Ikegami, A. Iwanami, M. Mukaino, A. Yasuda, S. Nori, Y. Mabuchi, H. Tada, S. Shibata, K. Saito, M. Matsushita, K. Kaibuchi, S. Okada, Y. Toyama, M. Nakamura, and H. Okano. 2011. Beneficial compaction of spinal cord lesion by migrating astrocytes through glycogen synthase kinase-3 inhibition. *EMBO Mol Med.* 3:682-696.
- Roberts, D.J., and S. Miyamoto. 2015. Hexokinase II integrates energy metabolism and cellular protection: Acting on mitochondria and TORCing to autophagy. *Cell Death Differ.* 22:248-257.
- Silver, J., and J.H. Miller. 2004. Regeneration beyond the glial scar. *Nat Rev Neurosci.* 5:146-156.
- Vandesompele, J., K. De Preter, F. Pattyn, B. Poppe, N. Van Roy, A. De Paepe, and F. Speleman. 2002. Accurate normalization of real-time quantitative RT-PCR data by geometric averaging of multiple internal control genes. *Genome Biol.* 3:RESEARCH0034.
- Zhang, Z., X. Deng, Y. Liu, Y. Liu, L. Sun, and F. Chen. 2019. PKM2, function and expression and regulation. *Cell Biosci.* 9:52.

February 17, 2023

RE: Life Science Alliance Manuscript #LSA-2022-01605-TR

Prof. Kyoungso Suk
Kyungpook National University School of Medicine
Department of Pharmacology
680 Gukchaebosang Street, Joong-gu
Daegu 41944
Korea, Republic of (South Korea)

Dear Dr. Suk,

Thank you for submitting your revised manuscript entitled "A multiplexed siRNA screen identifies key kinase signaling networks of brain glia". We would be happy to publish your paper in Life Science Alliance pending final revisions necessary to meet our formatting guidelines.

- please address Reviewer 2's remaining points
- please upload your supplementary figure files as single files and add the supplementary figure legends to the main manuscript text
- please add your video legend to the main manuscript text
- please add a category for your manuscript to our system
- please add the Twitter handle of your host institute/organization as well as your own or/and one of the authors in our system
- please use the [10 author names, et al.] format in your references (i.e. limit the author names to the first 10)
- you may want to consider uploading Figure 9 as a Graphical Abstract, rather than a figure. This is up to you.

Figure Check:

- please add scale bars to Figure 8B and C
- please add sizes next to to blots in Figure S3B

A. FINAL FILES:

B. MANUSCRIPT ORGANIZATION AND FORMATTING:

Sincerely,

Reviewer #1 (Comments to the Authors (Required)):

In the revised manuscript, authors have provided thorough responses to comments improving manuscript. I endorse it for publication.

Reviewer #2 (Comments to the Authors (Required)):

The authors have addressed most of the issues raised by this reviewer. However, some information on the statistics (as noted in the original review, #1) still seems to be unclear. In the methods, the authors state that data is given as mean +/- SD or mean +/- SEM. However, it is not specified in all data presentations, whether SD or SEM is shown. This e.g. relates to table S3 (but also to other data presentations throughout the MS as well). In table S3, three preparations are listed, but for each preparation a mean +/- error bar is given. This raises two questions: a) is the error SD or SEM? b) What is the single replicate which is used here for calculation for each preparation? Are these a number of images taken from the preparation (how many?), or what is it? Please clarify these issues on n numbers, data presentation etc throughout the MS.

Editor's comments:

-please address Reviewer 2's remaining points

Response: Reviewer 2's remaining points were addressed.

-please upload your supplementary figure files as single files and add the supplementary figure legends to the main manuscript text

Response: Done.

-please add your video legend to the main manuscript text

Response: Added as below

Movie. Time-lapse images of the wound healing assay. Representative time-lapse images of the wound healing assay, taken every 2 h for a total period of 48 h. *Left*, control MGCs; *middle*, MGCs incubated with Pdk2 inhibitor (AZD7545, 1 μ M); *right*, MGCs from *Pdk2* knockout (KO) mice.

-please add a category for your manuscript to our system

Response: Done in the system.

-please add the Twitter handle of your host institute/organization as well as your own or/and one of the authors in our system

Response: Done in the system.

-please use the [10 author names, et al.] format in your references (i.e. limit the author names to the first 10)

Response: Done.

-you may want to consider uploading Figure 9 as a Graphical Abstract, rather than a figure. This is up to you.

Response: Figure 9 uploaded as a Graphical Abstract, leaving the figure where it was.

Figure Check:

-please add scale bars to Figure 8B and C

Response: Added.

[Figure removed by LSA Editorial Staff per authors' request]

Figure 8. Role of pyruvate dehydrogenase kinase isoform 2 (Pdk2) in glial cell migration and glycolytic metabolism. (A) Simplified “Central carbon metabolism in cancer” pathway. Asterisks indicate the kinases associated with specific signaling pathways, identified in the siRNA screen. (B) Representative images from the wound healing assay with mixed glial cells (MGCs) from wild-type (WT) mice. The cells were incubated with a Pdk2 inhibitor (AZD7545, 1 μ M), glycolysis inhibitor (2-DG, 2 mM), or an OXPHOS inhibitor (oligomycin or FCCP, 1 μ g/mL) for 48 h. Scale bar = 100 μ m. (C) Representative images from the wound healing assay of MGCs from *Pdk2* knockout (KO) mice. The cells were incubated with the glycolysis inhibitor or OXPHOS inhibitor for 48 h as described in (B). Scale bar = 100 μ m.

-please add sizes next to blots in Figure S3B

Response: Added.

[Figure removed by LSA Editorial Staff per authors' request]

Reviewer #2:

Comment: The authors have addressed most of the issues raised by this reviewer. However, some information on the statistics (as noted in the original review, #1) still seems to be unclear. In the methods, the authors state that data is given as mean \pm SD or mean \pm SEM. However, it is not specified in all data presentations, whether SD or SEM is shown. This e.g. relates to table S3 (but also to other data presentations throughout the MS as well). In table S3, three preparations are listed, but for each preparation a mean \pm error bar is given. This raises two questions: a) is the error SD or SEM? b) What is the single replicate which is used here for calculation for each preparation? Are these a number of images taken from the preparation (how many?), or what is it? Please clarify these issues on n numbers, data presentation etc throughout the MS.

Response: In response to this comment, we clarified our description on the statistics throughout the manuscript (Figures 6, 7, S7, S8, and S9) (see below).

Also, with respect to table S3, here is our answer to the two questions raised by the reviewer.

a) Is the error SD or SEM?

→ The values are presented as the mean \pm SEM.

b) What is the single replicate which is used here for calculation for each preparation? Are these a number of images taken from the preparation (how many?), or what is it?

→ The single replicate of each preparation is the number of cell culture well. Mixed glial cells were seeded onto three wells (n=3 replicates) of a 96-well plate for each experimental group. Three to four images were taken per well to obtain an average value. The data shown for each preparation are mean \pm SEM of these three average values obtained from the three wells. We have added this information on the statistics in the revised Table S3 as below.

Table S3-1. Percentage of glial cell types in MGCs cultures.

^b Percentage of each cell type was calculated. Total number of DAPI-positive cells was set to 100%. The data shown for each preparation are mean \pm SEM obtained from three wells (n=3 replicates). Three to four images were taken per well to obtain an average value.

Table S3-3. Transfection efficiency in MGCs and interassay variation across different cultures.

^a Percentage of transfection efficiency was calculated as follows: (the number of transfected cells/the number of total cells stained with each cell marker) x 100. The data shown for each preparation are mean \pm SEM obtained from three wells (n=3 replicates). Three to four images were taken per well to obtain an average value.

Figure 6. *Itk* expression and activity in the brain. (A) Simplified “T cell receptor signaling” pathway. Asterisks indicate the kinases identified in the siRNA screen. **(B)** *Itk* mRNA levels

in MGC stimulated with LPS (1 $\mu\text{g}/\text{mL}$) for 6 h. Data are presented as mean \pm SEM (n = 3 replicate wells per group). The assay was repeated at least twice with similar results and the interassay coefficients of variation was <20%, indicating a high level of confidence in the result. (C) *Itk* mRNA levels in the inflamed brain 24 h after intraperitoneal LPS injection (5 mg/kg). *Itk* mRNA levels were measured by q-PCR in the whole brain. Data are presented as mean \pm SEM (n = 3 mice per group). (D) Immunofluorescence analysis of phosphorylated Itk (p-Itk) and total-Itk (Itk) in the hippocampus of LPS-induced inflamed brains. Scale bar = 25 μm . Mice were injected with LPS (5 mg/kg) intraperitoneally (i.p.). At 24 h after LPS injection, brains were prepared for immunofluorescence analysis. Glial cells were stained with anti-Iba-1 (microglia) and anti-GFAP (astrocytes) antibodies. A quantification of microglial cells (Iba-1⁺) co-stained with p-Itk (Iba-1⁺ p-Itk⁺) or Itk (Iba-1⁺ Itk⁺) is also shown (right). Data are presented as mean \pm SEM (PBS, n = 3 mice; LPS, n = 5 mice; each data point indicates the average values of six fields of view per mouse). (E) Immunoblots of kinases related to the T cell receptor signaling pathway. Primary microglia were treated with LPS (1 $\mu\text{g}/\text{mL}$). Total protein was harvested at the indicated time points and subjected to immunoblot analysis (upper) and quantification (lower). p-PLC- γ , phosphorylated PLC- γ ; PLC- γ , total PLC- γ ; p-Vav, phosphorylated Vav1; Vav1, total Vav1. Data are presented as mean \pm SEM (n = 3 replicate wells per group). Unpaired *t*-test (B, C, and D), One-way ANOVA followed by Tukey's post hoc test (E), **P* < 0.05. The assay was repeated at least twice with similar results and the interassay coefficients of variation was <20%, indicating a high level of confidence in the result.

Figure 7. Inhibition of microglial Itk alleviates LPS-induced neuroinflammation. (A) Interleukin-2-inducible T-cell kinase (Itk) inhibitors (BMS509744 and GSK2250665A) diminished LPS-induced NO production in primary microglial cells. The cells were stimulated with LPS (1 $\mu\text{g}/\text{mL}$) for 24 h. NO production was measured by nitrite concentration in the cultured media and cell viability was assessed using an MTT assay. Data are presented as mean \pm SEM (n = 3 replicate wells per group). (B) Immunoblots of PLC- γ (an Itk substrate) after treatment of primary microglia with LPS (1 $\mu\text{g}/\text{mL}$) in the presence or absence of the Itk inhibitor BMS509744 (1 $\mu\text{g}/\text{mL}$). The quantification is shown in the adjacent graph. Data are presented as mean \pm SEM (n = 3 replicate wells per group). (C) Immunohistochemistry of glial cells in the hippocampal region. Mice were intraperitoneally (i.p.) injected with LPS (5 mg/kg) with or without BMS509744 (10 mg/kg, i.p.). Astrocytes and microglia in the hippocampus were immunostained with anti-GFAP and anti-Iba-1 antibodies, respectively, at 24 h after LPS treatment. Scale bar = 400 μm . The quantification of glial activation is shown in the adjacent graph. Data are presented as mean \pm SEM (n = 5 mice per group; each data point indicates the average value of six fields of view per mouse). (D) Expression of proinflammatory genes (*Tnf- α* , *Il-1 β* , and *Nos2*) in mouse brains. After LPS injection (i.p., 24 h), hippocampal tissues were subjected to RT-PCR analysis. Data are presented as mean \pm SEM (n = 5 mice per group). (E) PLC- γ immunoblots in hippocampal tissues at 24 h after LPS injection (5 mg/kg, i.p.) with or without the Itk inhibitor BMS509744 (10 mg/kg, i.p.). The quantification is shown in the adjacent graph. Data are presented as mean \pm SEM (n = 3 mice per group). (F) Effect of Itk inhibition on LPS-induced behavioral impairments. After three days of LPS treatment (5 mg/kg, i.p. injection) with/without the Itk inhibitor BMS509744 (10 mg/kg, i.p.), spatial working memory (Y-maze test) and depression-like behavior (forced swim test) were measured. Data are presented as mean \pm SEM (n = 6 mice per group). The Y-maze spontaneous alternation test showed that impaired spatial memory following administration of LPS was reversed by BMS509744 injection (left panel). The forced swim test revealed that the significant increase in

immobility observed in LPS-injected mice was alleviated by BMS509744 injection (right panel). Two-way ANOVA followed by Tukey's post hoc test (A), One-way ANOVA followed by Tukey's post hoc test (B, E, and F), Unpaired *t* test (C and D). **P* < 0.05. The assays (A and B) were repeated at least twice with similar results and the interassay coefficients of variation was <20%, indicating a high level of confidence in the result.

Figure S7. Comparison of glial phenotypes upon stimulation with DAMPs and different neuroinflammatory aspects altered by kinase knockdown. (A) Comparison of glial nitric oxide (NO) production and migration after treatment with LPS or damage-associated molecular patterns (DAMPs). Mixed glial cell (MGC) cultures were transfected with 12 kinase siRNAs overnight. After 48 h, LPS (1 µg/mL), HMGB1 protein (20 µg/mL), or MGC lysates (1 µg/mL) were applied. (a) Comparison of NO production. **Data are presented as mean ± SEM (n = 3 replicate wells per group).** (b) Comparison of migration. Effects of kinase siRNA transfection in each assay compared with an untransfected control. **Data are presented as mean ± SEM (n = 3 replicate wells per group).** Abbreviation: ns, not significant. (B) Comparison of different neuroinflammatory aspects altered by kinase knockdown. At 14 days *in vitro*, MGC cultures were transfected with the 12 kinase siRNAs also used in the validation experiments in our study, overnight. After 48 h, LPS (1 µg/mL) was applied. (a) Screen dataset for the NO assay. **Data are presented as mean ± SEM (n = 3 replicate wells per group).** (b) Superoxide production assay. Transfected MGC cultures were stimulated with LPS (1 µg/mL) for 30 min and incubated with 10 µM dihydroethidium. The cells were then washed with ice-cold PBS and superoxide production was evaluated with a microreader (excitation: 534 nm; emission: 580 nm). **Data are presented as mean ± SEM (n = 3 replicate wells per group).** (c, d) TNF-α or MMP-9 protein release in the cultured media of transfected MGC cultures. The transfected cells were stimulated with LPS (1 µg/mL) for 48 h; then, culture media were harvested. Levels of (c) TNF-α or (d) MMP-9 protein were measured using a mouse TNF-alpha DuoSet ELISA Kit or mouse Total MMP-9 Quantikine ELISA Kit. **Data are presented as mean ± SEM (n = 3 replicate wells per group).** (e) Summary of the complete data. Abbreviations: ns, not significant. Values (A and B) are represented as fold change (log₂ FC) or not significantly different (ns) relative to the control (without siRNA). Red and blue indicate an increase or decrease, respectively, compared with each control. Unpaired *t* test. **P* < 0.05 vs. untransfected control. All assays were repeated at least twice with similar results and the interassay coefficients of variation was < 20%, indicating a high level of confidence in the result.

Figure S8. Comparison of glial phenotypes between MGC and single cell type culture. (A) Comparison of LPS-induced NO production among mixed glial cell, microglia, and astrocyte cultures after transfection with kinase siRNAs. At 14 days *in vitro*, astrocytes were isolated from MGC by shaking overnight to remove other cell types. Microglia were isolated by mild-trypsin treatment. Cells were transfected with 12 kinase siRNAs overnight. After 48 h, LPS (1 µg/mL) was applied. NO production was measured at 48 h after LPS treatment and compared to that of an untransfected control. (a) Screen dataset (NO production in MGC), (b) NO production in microglia, (c) NO production in astrocytes, and (d) summary of all data in the NO assay. **Data are presented as mean ± SEM (n = 3 replicate wells per group).** (B) Comparison of LPS-induced migration among mixed glial cell, microglia, and astrocyte cultures after transfection with kinase siRNAs. At 14 days *in vitro*, astrocytes were isolated

from MGCs by shaking overnight to remove other cell types. Microglia were isolated by mild-trypsin treatment. Cells were transfected with 12 kinase siRNAs overnight. After 48 h, LPS (1 $\mu\text{g}/\text{mL}$) was applied. (a) Screen dataset (migration in MGCs), (b) migration in microglia, (c) migration in astrocytes, and (d) summary of all data in the migration assay. Abbreviations: ns, not significant. Values (A and B) are represented as fold change ($\log_2 \text{FC}$) or not significantly different (ns) relative to the control (without siRNA). Red and blue indicate an increase or decrease, respectively, compared with each control. **Data are presented as mean \pm SEM (n = 3 replicate wells per group).** (C) Comparison of LPS-induced cell death among MGCs, microglia, and astrocyte cultures after transfection with kinase siRNAs. At 14 days *in vitro*, astrocytes were isolated from MGCs by shaking overnight to remove other cell types. Microglia were isolated by a mild-trypsin treatment. Cells were transfected with the kinase siRNAs overnight. After 48 h, LPS (1 $\mu\text{g}/\text{mL}$) was applied. Cytotoxicity was measured by lactate dehydrogenase assay at 48 h after LPS treatment. **Data are presented as mean \pm SEM (n = 3 replicate wells per group).** Unpaired t test. * $P < 0.05$ vs. untransfected control. All assays were repeated at least twice with similar results and the interassay coefficients of variation was $<20\%$, indicating a high level of confidence in the result.

Figure S9. Itk activity in the microglial cells. (A) Itk activity assay in BV-2 microglial cells. The microglial cells were incubated with LPS (100 ng/mL) and a serially diluted Itk inhibitor (BMS509744). Cell lysates were harvested in the presence of protease and phosphatase inhibitor. (a) Itk activity assay 1. We first determined the optimal amounts of cell lysate (containing Itk and other proteins) for measuring Itk activity based on signal-to-background ratio. A substrate (poly E_4Y_1)/ATP mix was added to the cell lysate and incubated for 1 h. After depleting the remaining ATP, Itk activity was visualized by kinase detection solution. The IC_{50} of BMS509744 was around 109 nM in this assay. **Data are presented as mean \pm SEM (n = 3 replicate wells per group).** (b) Itk activity assay 2. The cell lysate was subjected to western blotting for the detection of phospho-PLC-gamma (Tyr783), a direct substrate of Itk. Based on the inhibition of PLC-gamma phosphorylation, the IC_{50} of BMS509744 was around 407 nM in this assay. **Data are presented as mean \pm SEM (n = 3 replicate wells per group).** All assays were repeated at least twice with similar results and the interassay coefficients of variation was $<20\%$, indicating a high level of confidence in the result. (B) Schematic drawing depicting the putative role of Itk (asterisk) in microglial activation signaling.

February 22, 2023

RE: Life Science Alliance Manuscript #LSA-2022-01605-TRR

Prof. KyoungHo Suk
Kyungpook National University School of Medicine
Department of Pharmacology
680 Gukchaebosang Street, Joong-gu
Daegu 41944
Korea, Republic of (South Korea)

Dear Dr. Suk,

Thank you for submitting your Research Article entitled "A multiplexed siRNA screen identifies key kinase signaling networks of brain glia". It is a pleasure to let you know that your manuscript is now accepted for publication in Life Science Alliance. Congratulations on this interesting work.

DISTRIBUTION OF MATERIALS:

Again, congratulations on a very nice paper. I hope you found the review process to be constructive and are pleased with how the manuscript was handled editorially. We look forward to future exciting submissions from your lab.

Sincerely,
